# Diversity and ecological roles of hidden viral players in groundwater microbiomes

Akbar Adjie Pratama[1,2,3,4,5], Olga Pérez-Carrascal[1,2], Matthew B. Sullivan [3,4,5,6] ✉ & Kirsten Küsel [1,2,7] ✉

Groundwater ecosystems harbor diverse microbial communities adapted to energy-limited, light-deprived conditions, yet the role of viruses in these environments remains poorly understood. Here, we analyzed 1.24 terabases of metagenomic and metatranscriptomic data from seven wells in the Hainich Critical Zone Exploratory (CZE) to characterize groundwater viromes. We identified 257,252 viral operational taxonomic units (vOTUs) (≥ 5 kb), with 99% novel at order, family and genus levels against global ocean, freshwater and/or other publicly available datasets. In silico host predictions suggest that vOTUs primarily targeted Proteobacteria, Candidate Phyla Radiation (CPR) bacteria, and DPANN archaea, which reflects abundant and active groundwater microbial members. Patterns of virus-host abundance ratios, CRISPR-spacers, and prophage screening suggest the potential for multi-layer interactions involving CPR/DPANN lineages, their hosts, and viruses. Additionally, we identified 289 KEGG metabolic modules, 31.1% of which were targeted by 3378 vOTUs encoded auxiliary metabolic genes (AMGs) linked to carbon, nitrogen, and sulfur cycling. These findings provide a baseline for exploring how viruses influence microbial community dynamics, metabolic reprogramming and nutrient cycling in groundwater.

Viruses are the most abundant biological entities on Earth, with an estimated population of $10^{31}$ virus-like particles estimated on the planet, approximately an order of magnitude more than estimated for microbial cells[1,2]. Of these, $10^{30}$ are thought derived from marine environments with concentrations ranging from $10^6$ to $10^8$ per milliliter[3]. These marine viruses play critical roles, including lysing of cells (20–40% microbes per day time[1–3]), transferring genes from one host to another ($10^{29}$ genes transferred per day[4]), and metabolically reprogramming infected cells into entirely new entities (termed "virocells"[5]) that are biochemically distinct in their ecosystem inputs and outputs compared with their uninfected sister cells[6,7]. In aggregate, this leads to global biogeochemical cycling impacts through mechanisms such as the "viral shunt" (which recycles organic matter within surface waters) and "viral shuttle," (which enhances microbial aggregation and sinking) likely underpinning recent global-scale modeled findings that viruses, more so than prokaryotes or eukaryotes, best predict carbon export via the ocean biological carbon pump[8–10]. Ocean viruses' roles in microbial evolution can also be large with virus-modulated gene transfer even impacting gene encoding metabolic proteins, such as photosynthetic reaction centers for globally-dominant cyanobacteria[11]. Finally, with hundreds of thousands of ocean virus genomes now available and cataloged globally, there are tens of thousands of examples of virus-encoded auxiliary metabolic genes (AMGs) have been identified, underscoring the functional role of viruses in regulating biochemical cycles of carbon, nitrogen, and sulfur throughout the global ocean[5,6,12]. These ocean data

[1]Aquatic Geomicrobiology, Institute of Biodiversity, Ecology and Evolution, Friedrich Schiller University Jena, Jena, Germany. [2]Cluster of Excellence Balance of the Microverse, Friedrich Schiller University Jena, Jena, Germany. [3]Department of Microbiology, The Ohio State University, Columbus, OH, USA. [4]Center of Microbiome Science, The Ohio State University, Columbus, OH, USA. [5]National Science Foundation EMERGE Biology Integration Institute, Columbus, OH, USA. [6]Department of Civil, Environmental and Geodetic Engineering, The Ohio State University, Columbus, OH, USA. [7]Centre for Integrative Biodiversity Research (iDiv) Halle-Jena-Leipzig, Leipzig, Germany. ✉e-mail: sullivan.948@osu.edu; kirsten.kuesel@uni-jena.de

suggest that viruses can thus profoundly affect microbial evolution, and nutrient cycling. Such findings are also paralleled in surface freshwater environments such as rivers[13], and lakes[14].

Despite this progress, the diversity, function, and ecological impact of viruses in many other environments remain largely unknown. One such critical but understudied environment is groundwater, Earth's largest reservoir of unfrozen freshwater, which sustains terrestrial surface ecosystems and plays a crucial role in global biogeochemical cycles, including carbon[15–17], nitrogen[18,19], sulfur[20,21], phosphorus[22], and various metals[23]. Groundwater is a microbially dominated habitat, characterized by nutrient scarcity, low organic carbon availability, and the absence of light, which collectively impose strong selective pressures on resident microbial communities[24]. Chemolithoautotrophy, a primary mode of carbon fixation in these environments, occurs at rates largely comparable to photoautotrophic carbon fixation in sunlit oligotrophic ocean waters[15]. Groundwater harbors abundant ultra-small prokaryotes, including members of the Candidate Phyla Radiation (CPR) and DPANN superphylum archaea[25–27], which possess highly streamlined genomes and are thought to rely on episymbiotic or syntrophic interactions for survival[24,25,28,29]. These microorganisms, which can constitute more than 50% of some groundwater communities[30], likely play fundamental roles in subsurface biogeochemical cycling. Much less is known about their viruses, particularly those infecting archaea, which are thought to play important roles in shaping host populations and functions in natural systems[31]. Due to the scarcity of cultured representatives for CPR bacteria, DPANN archaea, and their viruses, cultivation-independent genomic approaches are one mechanism to uncover their biology, including infecting viruses, virus-encoded AMGs, and virus-host interactions. This knowledge gap is especially critical in the context of climate change, as declining groundwater levels due to rising temperatures and shifting precipitation patterns are expected to impact microbial dynamics and biogeochemical processes[32,33]. Understanding viral activity in these systems is essential for predicting how microbial interactions and nutrient cycles may respond to environmental change.

Here, we applied viral ecogenomic analyses that integrate ecological and genomic data to explore viral community structure and function across seven groundwater wells within the Hainich Critical Zone Exploratory (CZE) in Germany. This well-characterized model system spans a range of physicochemical gradients, from oxic to anoxic conditions[34], and hosts diverse microbial populations, including under-studied chemolithoautotrophs (e.g., *Nitrospiria* and *Sulfurifustaceae*) and ultra-small prokaryotes (e.g., CPR bacteria and DPANN archaea)[25–27]. We hypothesize that viruses shape groundwater microbiomes by infecting ecologically important microbial taxa and contributing critical functions, reflecting adaptations to the subsurface environments. By elucidating the diversity and functional roles of viruses in groundwater, this study provides a baseline toward understanding viral contributions to subsurface microbial evolution and ecosystem functioning in one of Earth's most expansive yet least explored habitats.

## Results and discussion
### Groundwater virus communities are diverse and unique
Our dataset is derived from a previous study of the Hainich CZE, which consisted of 0.515 terabases of short-read metagenomic data from 31 groundwater samples sampled in 2019 from six wells spanning diverse hydrochemical and geochemical conditions (Fig. 1a-c). To these, we added 0.725 terabases of metagenomic data from 34 samples collected three years apart from the same six wells as the previous study, plus one additional well (seven wells total) sampled to cover another well at the end of the hillslope (H53). This combined dataset totals 1.24 terabases across 65 samples. To improve the recovery of species-level viral operational taxonomic units[35] (vOTUs), we also added six long-

read metagenomes. Additionally, to assess DNA viruses' activity for the subset of taxa captured at this time point, we leveraged ~22 gigabases of metatranscriptomic data from 17 groundwater samples from five wells collected in 2015[21] (see detailed information on the sampling years and wells in Supplementary Data 1).

We first identified virus contigs and clustered these into vOTUs, using a threshold of ≥95% average nucleotide identity over 80% of the smallest contig[36]. Because our datasets were cellular-fraction metagenomes, we applied additional filtering steps to remove potential microbial sequences in our final set of vOTUs. This included assessing genomic context and removing vOTUs with genes potentially misidentified as viral, but more likely representing mobile genetic elements in cellular genomic islands (Supplementary Figs. 1, 2, see Methods). In total, the 65 short-read and 6 long-read metagenomes (Supplementary Data 1) resulted in 23,996,780 assembled contigs (≥1 kb), where 4,708,626 (19.62%) were identified as virus contigs. To assess genetic variability at the approximately species level, these ~4.7 M virus contigs were clustered into ~2 M vOTUs (≥1 kb), including 257,252 vOTUs (≥5 kb), of which 82,245 were ≥10 kb, respectively (Supplementary Data 2). Accumulation curves suggested that sampling was nearing, but not likely saturated (Supplementary Fig. 3a), indicating high, but under-sampled virus richness, even though these were deeply sequenced prokaryotic metagenomes.

Focusing on the larger vOTUs of our dataset (~82 K vOTUs, ≥10 kb), we advanced our understanding of viral diversity in groundwater ecosystems as follows. Compared to available groundwater data, the number of vOTUs in our study is 23 times the 3580 groundwater vOTUs cataloged in the IMG/VR database[37], 52.7 times and 59 times than of other groundwater virome studies New Zealand[38] (1571 vOTUs) and the southeastern coast of Sweden[39] (1407 vOTUs). That said, a recent large-scale groundwater survey (Wu et al.[40]) assembled 107,610 vOTUs ≥10 kb, ~1.3-fold more than in our study. Relative to other aquatic biomes where large-scale datasets are available, our single groundwater site captures total vOTUs 42% the size of that found in global ocean studies (Global Ocean Virus Database, version 2, 195,728 vOTUs)[9], 36% when also including the viruses in GOV2 plus those from the matched prokaryote-fraction metagenomes (230,944 vOTUs)[41]. Moreover, our vOTU count far exceeds the number reported for the deep oceanic trench (1628 vOTUs)[42], rivers (1230 vOTUs)[13], and lakes (156 vOTUs)[43].

To assess the taxonomy, we combined marker gene-based classification (geNomad)[44] with gene-sharing networks[45], but with updated analytics and the reference database RefSeq v230 (vConTACT3 to allow hierarchical taxonomy[46]). The geNomad, which relies on marker gene collections mapped to ICTV taxonomy (version 2022), confidently assigned ~80% of vOTUs to established taxa up to the phylum/class level, while the remainder remained unclassified, suggesting either novel lineages or under-sampled regions of viral sequence space (Supplementary Data 3). At finer ranks, where vConTACT3 provides robust resolution, novelty sharply increased: only 0.3% of vOTUs were assigned at the order level, and ≥99.8% at family and 100% at genus represented novel lineages. Together, these results indicate that while a large fraction of vOTUs belong to recognized viral realms and higher taxa, most diversity occurs in novel orders, families, and genera not yet represented in reference databases (see Supplementary Fig. 4 and Supplementary Data 3).

Next, we compared our dataset to public virome databases and conducted cross-biome analyses at the vOTUs-level (only those of ≥10 kb, 82,756 vOTUs; see Methods). The public databases used including IMG/VR v4.1[37] (groundwater biome), Gut Virome Database (GVD)[47], Global Soil Virome database (GSV)[48], Global Oceans Viromes 2.0 database (GOV2)[9], Viral RefSeq v.222[49], Fennoscandian Groundwater Virome (FGV)[39], Groundwater Virome Catalogue (GWVC)[40], and Rumen Virome Database (RVD)[50] (Supplementary Fig. 5). At the vOTU level, not a single sequence from any of these datasets clustered with

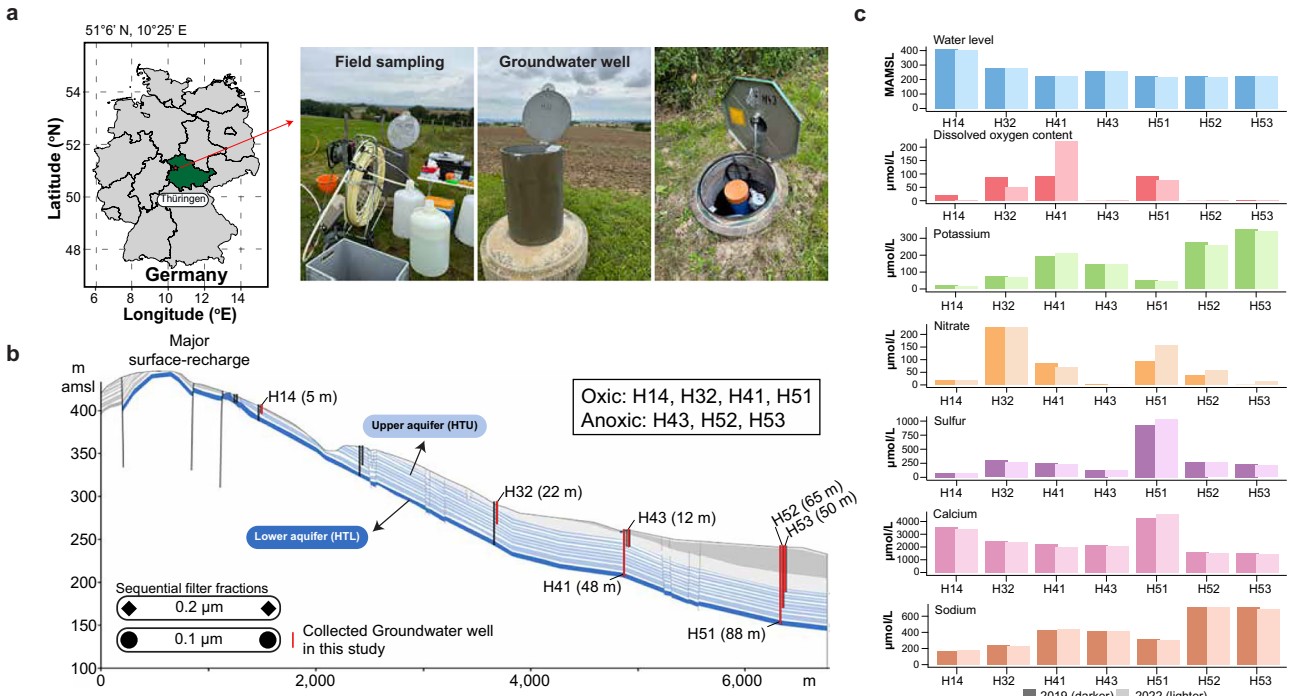

**Fig. 1 | Groundwater sampling design. a** Groundwater sampling was conducted in 2019 and 2022 at the Hainich Critical Zone Exploratory (CZE) located in Thuringia, Germany (indicated by the red star in the map to the left; 51°6′N, 10°25′E). Photographs show the field sampling equipment and a close-up of one of the groundwater wells. **b** A schematic cross-section of the Hainich CZE sampling site. Groundwater wells (represented by vertical red lines) were installed at various depths within a hillslope area. The groundwater monitoring transects span approximately 6 km. The multistorey aquifer system comprises a lower, karstified main aquifer (shown in blue, and labeled HTL for Hainich transect lower aquifer) and an upper mudstone-dominated aquifer (shown in teal, and labeled HTU for Hainich transect upper aquifer). Groundwater samples were collected using a sequential filtration through 0.2- (diamond symbol) and 0.1-μm (circle symbol) pore size filters. The figure is modified from Kohlhepp et al.[34] **c** Selected hydrochemical parameters of the groundwater of the seven wells are presented. For well H43, water level values were obtained from the two subsequent measurement campaigns, as no data were available during the metagenome sampling period. Complete and additional parameters are provided in Supplementary Data 7.

---

our groundwater viruses under applied thresholds and approaches, suggesting endemism of the groundwater viruses. We also compared our groundwater vOTUs to biome-specific vOTUs from other environments (freshwater, marine, and additional groundwater systems) using consistent thresholds (95% ANI, 50% and 80% coverages). These comparisons revealed similarly low levels of vOTU sharing not only between our data and other biomes, but also among non-groundwater datasets (e.g., ocean vs. river systems; Supplementary Fig. 6a, b). This suggests that limited vOTU overlap may reflect a broader pattern of viral biogeographic structure, where biome-specific endemism is common. Thus, while our results underscore the distinctiveness of groundwater viruses, they also indicate that low vOTU overlap is a general feature of viral community structure across ecosystems.

We next asked how many vOTUs were detectably active within the metatranscriptomes (17 samples), using previously established approaches and cut-offs[51], where viruses are considered active if at least one expressed gene was detected per 10 kb of genome[51] (see Methods). The results revealed that 23.57% (19,383) of vOTUs met this criterion (Supplementary Fig. 4b, Supplementary Fig. 7a, and Supplementary Fig. 8), with variation in activity between wells (p-value < 0.0001). Well H41 had the highest viral activity, with 32.6% of all identified vOTUs being active at this site (see Supplementary Data 4). To put this result into perspective, several studies using metatranscriptomic approaches have reported transcript activity of dsDNA viruses in aquatic systems, including both marine environments[52] and freshwater lakes[53]. And the handful of quantitative studies reported that ~58% of vOTUs were identified as active in permafrost soils, using a similar approach[51,54], and up to 73% in Arctic soils using stable isotope probing[55].

## Ecological patterns of the virus community are well-specific

Viruses can shape microbial turnover and adaptation in groundwater, but it is unclear whether they exhibit the well-specific structuring seen in microbes[28,30,56]. Determining this spatial partitioning is key to understanding how the aquifer environment impacts virus-host interactions. The Hainich CZE groundwater system is characterized by distinct hydrological and geochemical gradients driven by stratified aquifers (Fig. 1b). Wells H14, H32, H41, and H51 access a more connected lower aquifer with greater water exchange, whereas wells H52 and H53 tap into an upper aquifer that is hydrologically isolated by alternating carbonate (limestone) and siliciclastic (marlstone) layers that restrict vertical water flow[34]. This hydrogeological contrast is reflected in geochemical profiles: connected wells exhibit oxic conditions, while isolated wells range from suboxic to anoxic[56]. Long-term microbial surveys revealed that these differences strongly structure microbial communities[30,57]: isolated wells (H52, H53) were dominated by core bacterial taxa (55–65%), whereas connected wells had lower core fractions (9–29%) but higher diversity (Shannon's H = 5.2-5.9)[56]. Building on this framework, we hypothesized that viral communities would exhibit similar well-specific patterns. To test this, we assessed vOTU abundances (via metagenomic read mapping) across 65 samples from all seven wells (Supplementary Fig. 9a). We then classified vOTUs into the same categories used for microbes, using two complementary schemes: (i) unique (present in only one well) versus shared (found in multiple wells), and (ii) core (present in >50% of wells), common (21–50%), and rare (1–20%). Overall, 58.2% of vOTUs were unique to a single well, 39.7% were shared, and 2.04% could not be categorized. According to the second category, 6.10% were core, 33.6% common, and 58.2% rare vOTUs, with 2.04% again uncategorized

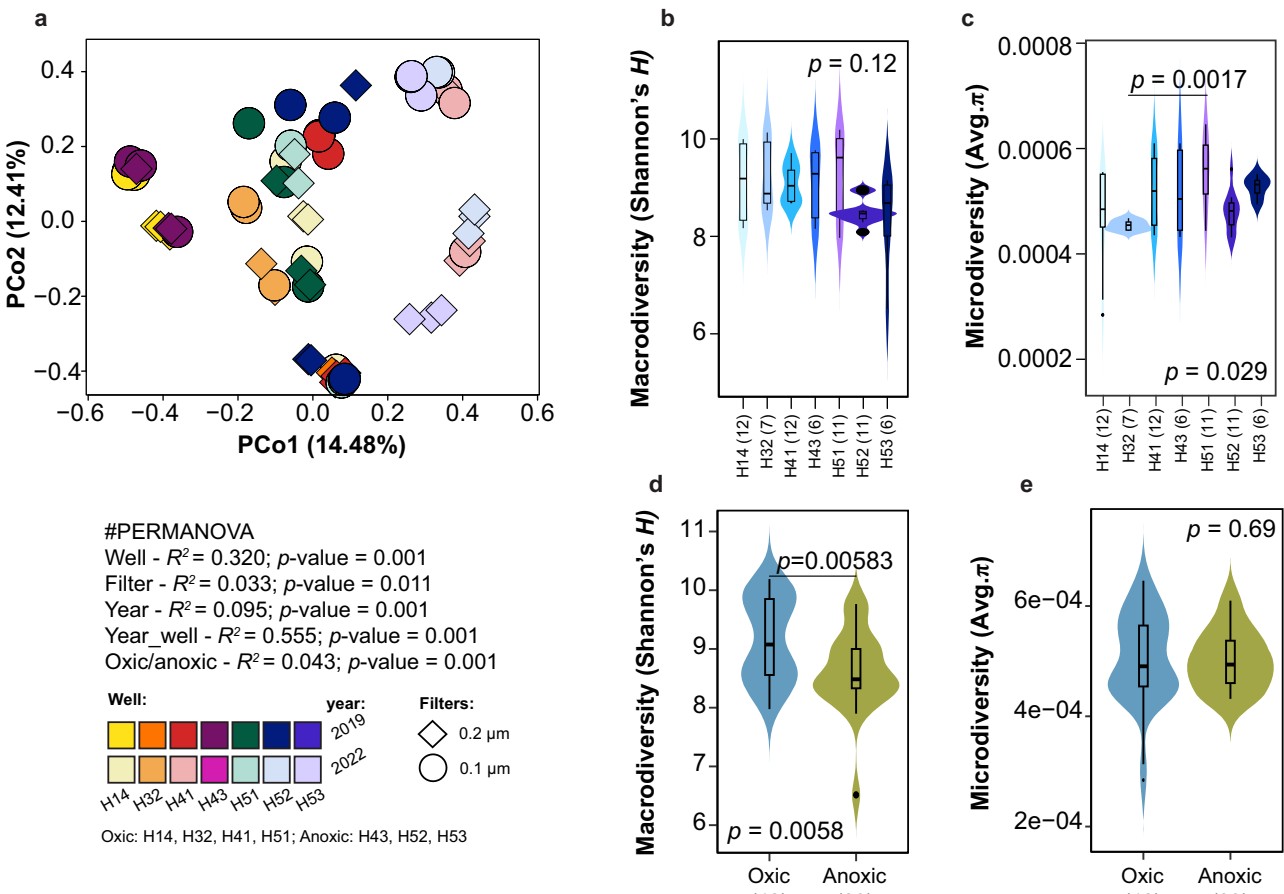

**Fig. 2 | Virus community patterns along the groundwater monitoring transect.**
**a** A principal coordinate analysis (PCoA) of Bray-Curtis dissimilarities among the groundwater viruses' communities was used to assess community structure. This analysis was based on the relative abundance of 257,252 viral operational taxonomic units (vOTUs; ≥5 kb), with each point plotted representing one of the 65 sampled metagenomes, color-coded by the well of origin. PERMANOVA was applied to test for significant differences in viral community composition among wells using Bray–Curtis dissimilarity. (**b, c**) Macrodiversity (Shannon's *H*) and microdiversity (Average π) across the wells. All pairwise comparisons are displayed, with statistically significances (*$p < 0.05$) indicated by connecting bars, and were conducted using the two-sided Kruskal-Wallis test. **d, e** Macrodiversity (Shannon's *H*), and microdiversity (Average π) in oxic versus anoxic wells. Significant differences between the two groups (**$p < 0.01$) are indicated by connecting bars. Comparisons were analyzed using the two-sided Mann-Whitney *U* test. **b–e** are based on independent biological replicates, where each point represents a single groundwater metagenomic sample collected from an individual well at a specific timepoint. Numbers in parenthesis show the sample sizes: **b, c** H14 (12 metagenomics samples), H32 (7), H41 (12), H43 (11), H52 (11), H53 (6). **d, e** Oxic wells (H14, H32, H41, H51): 42 metagenomics samples; Anoxic wells (H43, H52, H53): 23 metagenomics samples. Violin/box plot shows, the center line shows the median; the bounds of the box represent the 25th and 75th percentiles; whiskers extend to 1.5× the interquartile range; points outside this range represent outliers.

(Supplementary Fig. 9b). The highest percentage of unique vOTUs (19.2%) was observed in the uphill, oxic shallow well H14, while H52 and H53, at the bottom of the hillslope, had the highest proportions of shared, core, and common vOTUs (Supplementary Fig. 9b, c). To further explore patterns in virus community structure, we used non-metric multidimensional scaling (NMDS) and principal coordinates analysis (PCoA) based on vOTU relative abundance table (Fig. 2a and Supplementary Data 5). Ordination analysis revealed that virus communities were structured primarily by well and year (year_well; PERMANOVA, $R^2 = 0.555$, $p < 0.001$). Other individual factors, such as well location, sampling year, filter size, and oxic/anoxic conditions, explained weaker variation (Supplementary Fig. 10a). Together, these results indicate that viral communities are structured by spatiotemporal context within individual wells, likely reflecting shifts in host composition and local environmental conditions over time.

To better characterize viral diversity in groundwater, we examined it at two levels: macrodiversity (inter-population diversity, via Shannon's *H*) and microdiversity (intra-population genetic variation, via average nucleotide diversity, π) (Supplementary Data 6). We first assessed macrodiversity across the groundwater wells and sampling

years. Shannon's *H* values showed no statistically significant differences between wells (Kruskal-Wallis $p = 0.12$; Fig. 2b), nor across years (Wilcoxon test, $p = 0.10$; Supplementary Fig. 11), suggesting relatively stable overall viral population over both space and time. This temporal consistency aligns with the broader NMDS/PCoA-based findings that viral community structure varies more notably within wells over time (year_well), rather than by year alone (Fig. 2a, Supplementary Fig. 10a). However, when grouping wells by redox regime, a significant difference in Shannon's *H* emerged between oxic and anoxic conditions (Wilcoxon test, $p < 0.01$; Fig. 2d). We next assessed microdiversity by calculating average nucleotide diversity (π), which captures within-population genetic variation. Unlike macrodiversity, microdiversity varied significantly across wells (Kruskal-Wallis, $p < 0.01$; Fig. 2c, also see Supplementary Figs. 12, 13), indicating that local environmental or biotic conditions may subtly shape viral genetic structure even when overall community composition remains stable.

Together, these findings suggest that macro- and microdiversity are shaped by distinct ecological filters. The observed redox-dependent variation in macrodiversity likely reflects host availability and activity (Supplementary Figs. 14–S16), for instance, oxic wells

harbored more diverse, active communities, with significantly higher microbial Shannon's $H$ (Wilcoxon $p < 0.01$; Supplementary Fig. 15i) and ~2.25-fold greater transcriptomic activity (Supplementary Fig. 16b), which may promote greater viral diversification. In contrast, micro-diversity patterns likely stem from local selective pressures that promote isolation-by-environment[9], including hydrological and geochemical barriers (Fig. 1c) that limit dispersal, and virus-host interactions (see virus-host ratio patterns later in this section) that drive within-population divergence even when community-level diversity remains constant[58]. While spatial partitioning highlights how viral communities differ between wells, it does not explain why these patterns arise. To address this, we next examined the geochemical and nutrient factors that may drive viral diversity and microbial community structure.

## Ecological drivers of the groundwater viruses

To assess the ecological drivers of virus communities, we analyzed 18 different geochemical and nutrient parameters from existing datasets (Supplementary Data 7; see Methods). Mantel analysis revealed that temperature significantly correlated with virus richness (Mantel's $p = 0.01–0.05$), while ammonium and potassium concentrations (Mantel's $p < 0.01$) were significantly correlated with virus macro-diversity (Supplementary Fig. 17). Host richness and Shannon's $H$ were significantly correlated with water level, pH, redox potential, base neutralizing capacity (acidity), and magnesium concentrations (Supplementary Fig. 17). Redox conditions, a known driver of microbial[28,56] communities, as also shown by our results for microbial MAGs and viral communities in groundwater (Fig. 2d, e; Supplementary Fig. 15i-n), appears to influence these threshold-dependent manner. Viral communities showed weak but significant grouping by redox state (oxic vs. anoxic wells; NMDS/PCoA, Fig. 2a, Supplementary Fig. 10a), and Shannon's $H$ was significantly higher in oxic wells ($p < 0.01$; Fig. 2d). However, virus Shannon's $H$ and richness showed no correlation with dissolved oxygen concentrations was detected (Mantel's $p$-value) (Supplementary Fig. 17). This suggests that viral diversity shifts are triggered by redox thresholds rather than gradual oxygen changes. In our dataset, microbial communities also showed higher diversity and distinct clustering under oxic conditions (Supplementary Fig. 15i-n). Similar oxygen-driven shifts in viral community structure have been observed in marine oxygen minimum zones, where oxygen concentration strongly correlates with viral composition and diversity[59]. Together, these results imply that oxygen availability may act as a categorical ecological switch influencing virus–host dynamics, rather than a factor that can be captured by oxygen measurements alone. While the local geochemical and nutrient data influenced the overall virus community composition (macrodiversity), none were found to significantly impact microdiversity (Supplementary Fig. 17). These geochemical and nutrient factors may, however, drive virus macro-diversity indirectly by altering viral replication rates, host availability, and selective pressures. For example, potassium nitrate, and chloride were previously identified as key drivers of virus community in a separate groundwater study[38], highlighting the role of geochemistry in structuring viral assemblages.

## Groundwater viruses are predicted to infect diverse prokaryotic phyla

Defining the potential host of viruses is a key to understanding their role in microbial community dynamics and ecosystem functions. To predict potential virus-host associations in groundwater, we used an integrative in silico approach that leverages multiple sequence-based features and a custom database augmented with 1275 metagenome-assembled genomes (MAGs) co-sampled from these wells (see Methods). The taxonomic composition of the 1275 MAGs used for host prediction was dominated by Patescibacteria (59.4% of total MAG relative abundance), followed by Nanoarchaeota (8.2%),

Proteobacteria (7.0%), Thermoproteota (1.4%), and other taxa contributing less than 1% (Supplementary Fig. 14). We identified virus–host associations (at a high host confidence score ≥90%) for 11.6% (9,615) of vOTUs, linking them to microbial MAGs spanning 60 phyla, 143 classes, 339 orders, and 650 families (Supplementary Fig. 18 and Supplementary Data 8). Among these identified associations (11.6%), Proteobacteria accounted for the highest proportion of virus-host linkages (21%), followed by Patescibacteria (18%), even though Proteobacteria represented only ~7.0% of the cumulative relative abundance of all MAGs (Fig. 3a, b, Supplementary Fig. 18b). This is in contrast with marine ecosystems where the highest virus linkages are to the most dominant taxa[60]. Assuming this is not a database artifact (i.e., poorly represented Patescibacteria), then this suggests that viral targeting is not solely driven by host abundance but may instead reflect host activity or physiological state, with Proteobacteria among the most transcriptionally active taxa in this system (Supplementary Fig. 16). Supporting this, metatranscriptomic analysis revealed that some of the most transcriptionally active MAGs belonged to Nitrospirota, Patescibacteria, and Proteobacteria (Supplementary Fig. 16), and viruses predicted to infect 52 phyla, with Proteobacteria exhibited the highest relative transcript abundance across wells (Supplementary Fig. 8). Together, this indicates that viral activity is shaped not only by host abundance but also by host transcriptional output, with highly active Proteobacteria potentially serving as especially productive viral hosts. These virus-host associations likely represent conservative estimates. Current in silico host prediction approaches, i.e., iPHoP, typically resolve hosts only at the genus level; iPHoP does not capture strain-specific linkages, which are highly variable and underrepresented in many taxa. Consequently, ~89% of vOTUs remain unlinked to hosts, largely due to incomplete viral genomes, limited representation of many microbial clades in host databases, and conservative prediction thresholds to maintain low false discovery rates. If some of these unassigned vOTUs infect underrepresented lineages such as CPR/DPANN or poorly characterized phyla, this would bias interpretations of viral dynamics. Benchmarking of iPHoP also reported that prediction success drops sharply for novel or underrepresented taxa; for instance, more than 70% of Patescibacteria genera remain without any predicted viral associations due to limited ref. 61. Additionally, potential database biases, such as the relative overrepresentation of Proteobacteria, may further skew observed virus-host linkages.

Next, we quantify viral pressure across microbial lineages. We calculated lineage-specific virus/host abundance ratios (VHRs), as a proxy for viral impact[51,54]. VHRs were calculated from the ratio of average per-base-pair coverage of viral contigs to that of their predicted host MAGs, normalized by sample sequencing depth[51]. This analysis was limited to virus-host pairs, meaning the results reflect only a subset of the total virus-microbe interactions in the system. Among the 35 lineages analyzed, 48.6% (17 out of 35) had VHRs greater than 1 (log10 scale), indicating that viruses outnumbered their respective hosts. The highest VHRs were observed in Actinobacteria, Cyanobacteria, Bacteroidota, and Proteobacteria (Fig. 3c; Supplementary Data 9), suggesting virus control over these microbial groups. These VHRs also varied across wells (Supplementary Data 9), reflecting environmental heterogeneity (Supplementary Figs. 19, 20). In oxic wells (H14, H32, H41, H51), Proteobacteria exhibited elevated VHRs, consistent with higher viral pressure in oxygen-rich environments that promote microbial growth. Conversely, Patescibacteria showed higher VHRs in anoxic wells, likely driven by distinct viral-host interactions adapted to low-oxygen conditions. These lineage-specific patterns mirror known metabolic strategies: Proteobacteria thrive in nutrient-rich, oxic conditions, whereas Patescibacteria specialize in anoxic environments[25]. These findings aligned with a recent study from lake sediment ecosystems that reported VHRs correlate with changes in microbial productivity and biogeochemical function[43]. More broadly, recent work also highlights that VHRs can vary widely across

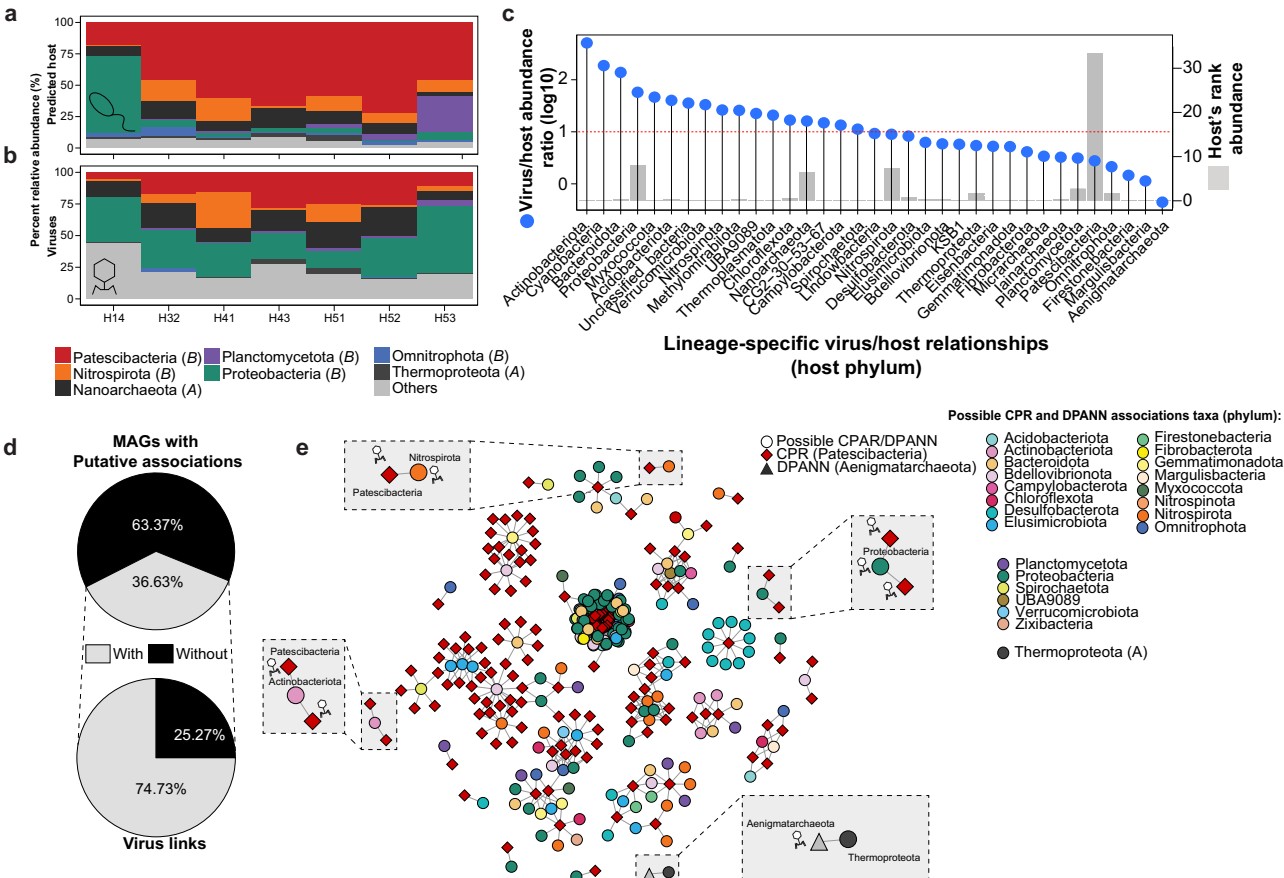

**Fig. 3 | Virus operational taxonomic unit-level analysis of in silico host prediction and multi-layer interactions CPR/DPANN–virus–host relationships.**
**a** Relative abundance (%) of predicted prokaryote hosts across seven wells, with panel (*A*) representing archaea and (*B*) representing bacteria. The overall prokaryote relative abundance is shown in Supplementary Fig. 14. Colors represent different phyla; "Others" includes all phyla with <1 % the total relative abundance (%).
**b** Relative abundance (%) of viruses, colored by the phylum of their predicted hosts.
**c** Lollipop plots depict virus:host abundance ratios (VHRs) by host lineage. VHR was calculated as the ratio of average per-base-pair coverage depth from read mapping to viral contigs versus host population genomes, normalized by sequencing depth in each sample. Dots represent mean VHR across the 65 metagenomes. The horizontal red dashed line indicates a 1:1 virus:host abundance ratio. Accompanying bar plots show host rank abundance based on mean VHR, ordered from highest to lowest. This phylum order is consistent with that used in Supplementary Fig. 20 to facilitate cross-panel comparisons. **d** Pie charts illustrate the overall number of metagenome-assembled genomes (MAGs) associated with other bacterial and archaeal MAGs (top) and, among these, the proportion linked to viruses (bottom). **e** Co-occurrence network showing potential association between CPR/DPANN and other bacterial and archaeal MAGs based on the 2019 sampling year. Co-occurrence analysis for the 2022 sampling year is shown in Supplementary Fig. 22. Circles represent potential host MAGs of CPR and DPANN. Red diamonds denote CPR MAGs, and dark-gray triangles represent DPANN MAGs. Zoomed-in clusters highlight specific associations between CPR/DPANN MAGs and putative host MAGs, such as Chloroflexota, and Proteobacteria. Virus links included in the inserted figure were inferred through in silico host prediction. Pie charts (**d**) show the percentage of MAG (1275) with CPR/DPANN associations based on the co-occurrence network analysis.

---

ecosystems, and lineage-specific VHRs offer higher resolution for probing virus-host interactions in complex microbiomes[62].

**Potential multi-layer interactions in the groundwater ecosystem**
Groundwater ecosystems harbor ultrasmall CPR bacteria and DPANN archaea, which often form close associations with other microbes[25–27,30,57]. These lineages have also been shown to be susceptible to viral infection[40,63], and recent work in hypersaline systems has demonstrated multi-layered "nested interactions" among DPANN archaea, their archaeal hosts, and associated viruses[64]. Similar multi-partner relationships involving CPR, their methylotrophic proteobacterial hosts, and jumbo phages have been reported in freshwater ecosystems[65]. Building on these emerging observations, we used genome-based inference to assess whether the groundwater communities exhibit similar interactions. To investigate this, we combined four analyses, including (*i*) virus-host prediction based on sequence similarity and host-matching approaches, (*ii*) MAG co-occurrence inferred from genome-resolved co-occurrence across samples, (*iii*)

prophage detection across the MAGs, and (*iv*) CRISPR spacer screening targeting the virus genome. First, in silico virus-host prediction analysis linked vOTUs to CPR and DPANN lineages (Supplementary Fig. 21). Among the vOTUs with host predictions (11.62% of the total), we found vOTUs infecting Patescibacteria (CPR), including the class-level numbers of associations for Paceibacteria had the highest number of associations (1001), followed by Microgenomatia (343), ABY1 (209), and Gracilibacteria (126). For DPANN, Nanoarchaea showed the most associations (1759) (Supplementary Fig. 21d, e). Previous groundwater virome studies reported 18 vOTUs linked to CPR and one vOTU linked to Nanoarchaeia in one study[38] and 97 non-redundant DPANN viruses were linked to 8 DPANN phyla in a separate study[66]. Our findings reported expanded putative linkages to these groups.

Second, we inferred host-symbiont interactions from co-occurrence network analysis based on MAG-based normalized relative abundance and third, assessed the fractions that were proviruses. To account for temporal confounders[67], co-occurrence analyses were performed separately for each year (Fig. 3d, e, and Supplementary

Fig. 22) and cautiously recognized that co-occurrence alone does not establish direct interactions, as shared environmental preferences may drive similar patterns. Still, as one line of evidence, it provides testable hypotheses for future experimental evaluations once new targeted toolkits become available. Given these caveats, our analysis suggests that 36.6% (467/1,275) of MAGs co-associated with other MAGs (Fig. 3d), of which 212 CPR MAGs were associated with non-CPR bacteria and 14 DPANN MAGs were associated with other non-DPANN archaea. We interpret these to suggest potential episymbiotic relationships (between CPR/DPANN-their potential microbial hosts). For example, Patescibacteria co-occurred with Chloroflexota, Desulfobacterota, Nitrospirota, Proteobacteria, and Omnitrophota, while Nanoarchaeota co-occurred with Iainarchaeota, all of these associations are also reported in other groundwater studies[27,57,68]. Of the MAGs with such associations, 74.7% (349/467) were also linked to vOTUs, supporting the possibility of concurrent host-symbiont-virus interactions. Separate from this, we also detected proviruses in 20.4% (106/519) of Patescibacteria MAGs and 29% (38/131) of Nanoarchaeota MAGs (Supplementary Fig. 23), indicating possible past or ongoing viral integration. This suggests that viruses are not only external pressures but also become embedded within these lineages through lysogeny.

Lastly, to further probe these interactions, we assessed viral pressure via virus-host abundance ratios (VHRs). Despite high abundance (59.4%; Fig. 3a) and activity (Supplementary Fig. 6 and Supplementary Fig. 16), VHR for Patescibacteria exhibited VHRs <1, contrasting with lineages like Proteobacteria or Bacteroidota. This likely reflects constrained infection dynamics due to lysogeny, low burst size, physical barriers from episymbiosis, or genomic streamlining limiting replication[25]. Notably, 18.4% of assigned vOTUs and 23.6% of active viruses linked to Patescibacteria (Supplementary Fig. 8a) indicate frequent viral encounters, implying a potential virus decoy role[69] or virus bait strategies[40] that have been proposed previously. To test whether this strategy/the same virus could infect CPR/DPANN and their possible hosts, we screened CRISPR spacers and revealed two vOTUs spacers linked to Desulfobacteriota and Patescibacteria (Paceibacteria), but no vOTU spacers linked to archaea and DPANN (Supplementary Data 10). While genetic code variation could theoretically enable a broad host range[40,63], our data lack direct evidence.

Together, our analyses suggest that viruses can interact with both CPR/DPANN and their putative microbial hosts, indicating multi-layer virus–symbiont–host relationships in groundwater. Most linkages point to distinct viral populations infecting CPR/DPANN versus hosts, while two CRISPR spacer matches suggest rare cross-host infection by the same vOTU, specifically between CPR and their putative bacterial host. Viruses that target the CPR/DPANN organisms may act as hyperparasites[64], controlling their population burden. Conversely, viruses infecting the microbial hosts could indirectly restructure or destabilize the entire partnership. Viruses that dual-target both partners could pose top-down pressure to both partners simultaneously. In the next section, we further show that viruses linked to CPR and DPANN carry AMGs that may modulate attachment and interaction dynamics. Our results provide testable hypotheses about multi-partner viral relationships that require experimental validation or higher-level resolution of genomic data, for example, isolates, single-cell or long-read sequencing.

### Viruses' influence on biogeochemical processes in groundwater
Viruses can affect ecosystem functioning by carrying host-derived auxiliary metabolic genes (AMGs) that modulate microbial metabolism during infection[51,70,71]. In oligotrophic groundwater, where both efficient nutrient uptake and utilization as well as biomass recycling are critical, we tested whether AMGs are both prevalent and expressed in situ. We identified putative AMGs from 82,245 vOTUs (≥10 kb) using DRAM-v[72] with adaptations from a recently developed method[73] (see

Methods). This yielded 4093 putative AMGs in 3378 vOTUs (4.1%) (Fig. 4a, b), grouped into 377 protein clusters (Supplementary Data 12). Functions spanned transport, and central carbon, amino acid, nitrogen, and sulfur metabolism (Supplementary Fig. 24). A large fraction overlapped with AMGs reported from the global ocean (75.6%)[41] and from a 20-year freshwater lake virome study (58.8%)[74] (Supplementary Fig. 24b), underscoring functional convergence across aquatic systems. AMG-encoding viruses were linked to hosts from 24 phyla, 44 classes, 75 orders, and 122 families (Supplementary Data 12), suggesting that AMG acquisition is a widespread viral strategy.

Next, we analyzed metatranscriptomic data to assess transcriptional activity. Of all expressed viral genes, 8.0% (686) corresponded to putative AMGs (Fig. 4c). Transcribed AMGs were associated with pathways such as the TCA cycle, glycolysis, and nitrogen and sulfur metabolism, pointing to viral involvement in shaping microbial energy flow. Transcript levels increased in downstream located wells H51 and H52, which showed elevated expression of genes related to peptidases (-45.6% of normalized expression per AMG category), amino acid metabolism (-14.4%), pyruvate metabolism (-9.3%), and electron transport (-9%) (Fig. 4c). This trend suggests that viruses increasingly modulate host metabolism as microbial communities stabilize and core taxa become dominant, particularly in H52, which is characterized by hydrologically isolated conditions[56]. Similar AMG functions have been reported in other nutrient-limited systems, such as the deep ocean[75], where viruses are thought to redirect host metabolism toward replication. While our data demonstrate in situ expression of AMGs (Fig. 4c, Supplementary Fig. 24), confirming their biochemical impact will require complementary approaches, such as stable isotope probing, metabolomics, or host-targeted proteomics.

Finally, we examined AMGs related to sulfur, nitrogen, carbon, and methane metabolism (Supplementary Figs. 24, 25). Sulfur-cycling AMGs (*cysA, cysH, cysE*) were linked to vOTUs predicted to infect Proteobacteria, Nitrospirota, and Desulfobacterota, taxa previously reported as key sulfate reducers in Hainich groundwater[18,21]. These AMGs mediate sulfur assimilation and organosulfur transformations and have been detected across diverse biomes[76]. Nitrogen-related AMGs (e.g., *nrfA*) were linked to Bdellovibrionota and detected only in well H41 (Fig. 4b), where elevated nitrification has been reported[19]. Putative AMGs associated with central carbon and methane metabolism were also identified. For instance, *cofE* (coenzyme F$_{420}$ biosynthesis) may influence methane flux, while AMGs linked to the TCA cycle and the reductive Wood-Ljungdahl pathway were associated with Bacteroidota, a group typically heterotrophic but occasionally carrying gene related to $CO_2$-utilizating pathways[15]. Although groundwater contributes only -0.2% of global methane emissions[77], the detection of these AMGs suggests that viruses could influence not only methane dynamics but also broader microbial carbon and nutrient cycling in these resource-limited ecosystems.

### AMGs encoded by viruses infecting CPR bacteria and DPANN archaea
CPR bacteria and DPANN archaea are among the most abundant taxa in Hainich groundwater, yet their streamlined genomes constrain metabolic capacity. To assess whether viruses might supplement these lineages, we examined AMGs encoded by their predicted viruses. Among CPR-associated viruses, 7.1% (125 vOTUs) carried AMGs related to carbon metabolism (e.g., pentose phosphate pathway, glycolysis), amino acid synthesis, and nutrient transporters (Supplementary Fig. 26; Supplementary Data 12). These AMGs may complement hosts' primarily fermentative, anaerobic metabolism, for example, glycolysis-related AMGs may enhance substrate-level phosphorylation under energy-limited conditions[75]. For DPANN-associated viruses, 5.8% (105 vOTUs) encoded AMGs involved in carbon utilization (e.g., pyruvate metabolism), sulfur reduction, and organic nitrogen processing. Putative functions included pyruvate dehydrogenase and threonine/

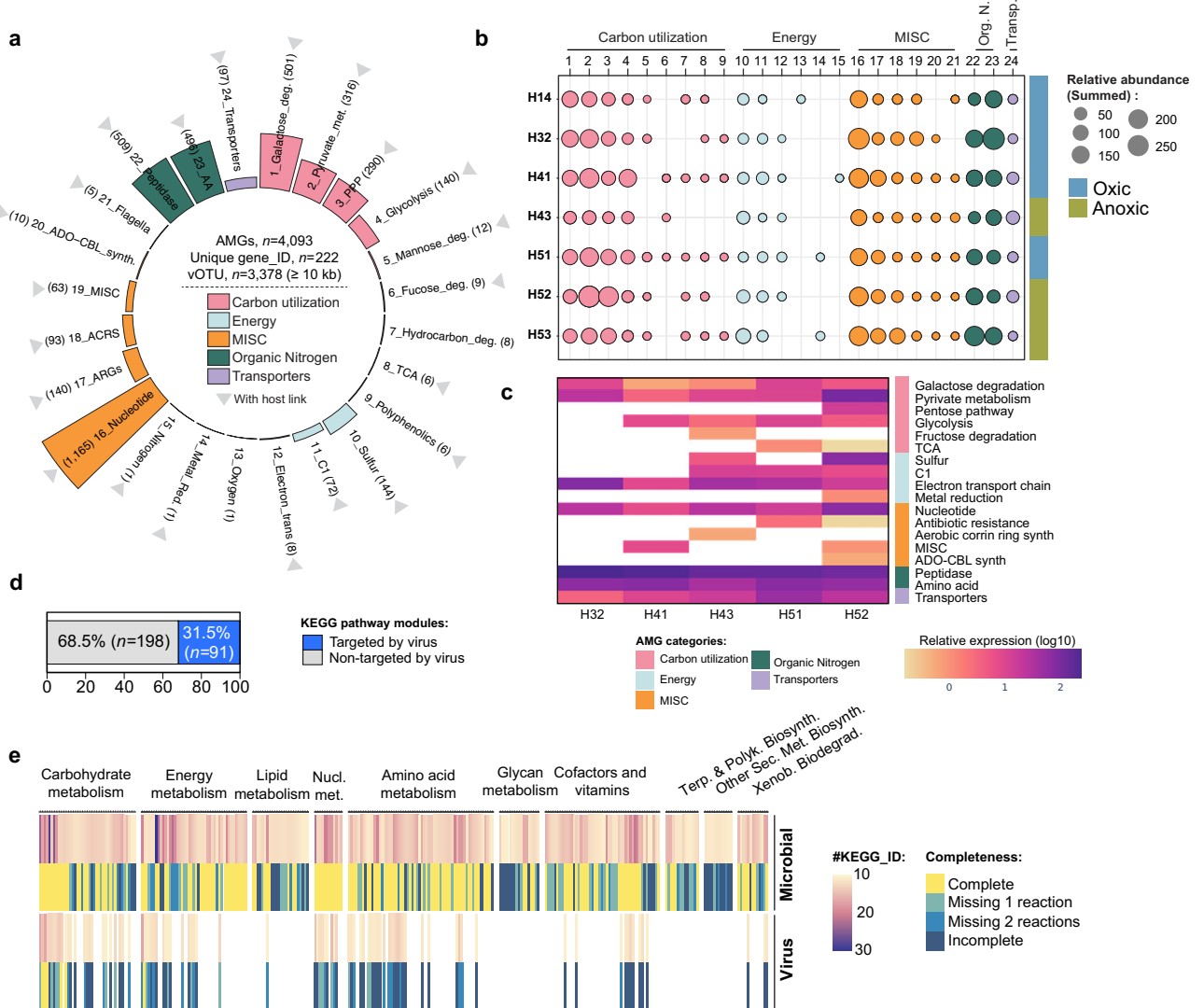

**Fig. 4 | The distribution and metatranscriptomic analyses of the putative auxiliary metabolic genes (AMGs). a** A circular bar plot depicts the identified auxiliary metabolic genes (AMGs), colored by functional category. Gray triangles indicate cases where host linkages were identified. **b** A bubble plot shows the distribution of viruses carrying AMGs across different wells, categorized by AMG type. Bubble size corresponds to the relative abundance of virus-carrying AMG. Oxic and anoxic wells are indicated by blue-sky and olive-green colors, respectively. **c** Heatmap showing the expression levels of the AMGs, expressed as relative transcript abundance in log10. Color intensity reflects the relative expression level of each viral AMG within a given metatranscriptome sample. Higher relative expression (log10) corresponds to more intense colors. **d** A stacked bar plot depicting the percentage of KEGG pathway modules that are targeted (blue), or not-targeted (gray) by viruses. **e** Heatmap representing reconstruction of microbial metabolic pathways for microbial (top) and putative AMGs (bottom). Colors reflect the number of KEGG IDs detected, and pathway completeness is indicated. Terp. & polyk. Biosynth. stands for terpenoid and polyketide biosynthesis, Sec.Met. Biosynth. = for secondary metabolite biosynthesis, and Xenob.biodegrad. = for Xenobiotic Biodegradation. The detailed metabolic reconstruction is provided in Supplementary Data 13.

serine dehydratases, which may redirect host carbon flow toward viral replication at the expense of host efficiency. Additionally, some AMGs were linked to cell surface modification; while not central to host metabolism, such functions may influence virus-host attachment and interactions[65].

At the community level, viruses encoding AMGs targeting 31.5% (91) of host KEGG metabolic modules, including the citrate cycle and cobalamin biosynthesis (Fig. 4d, e, Supplementary Data 13). For comparison, a global marine virome study estimated that 19% of viral populations carry AMGs (after adjusting for genome fragmentation, with ~9% directly observed in fragmented metagenomes)[41]. Despite differences in prevalence, the breadth of targeted pathways was similar between groundwater (31.5%) and marine systems (37.6%), suggesting that metabolic reprogramming via AMGs is a convergent viral strategy in oligotrophic environments.

Overall, this study explores viral roles in shaping the groundwater microbiome and subsurface biogeochemical processes. We uncovered a vast and largely uncharacterized viral diversity, with 100% representing novel genera, including a number of CPR and DPANN-associated viruses that encode diverse AMGs of interest and expand virus genomic representation of these hosts at least an order of magnitude. These findings suggest that viral interactions within the groundwater microbiome are more abundant and extensive than previously recognized, with potential consequences for microbial adaptation and ecosystem function. To explore this, we focus on potential multi-layer interactions among CPR/DPANN lineages, their hosts, and viruses, to highlight viruses as players within host-symbiont networks in groundwater microbiomes. The virus-encoded AMGs invoke links to carbon, nitrogen, and sulfur cycles where virus infection could presumably then target and modulate the key host

metabolisms that underpin these nutrient cycles in energy-limited environments. Although our analyses focus on dsDNA viruses, RNA viruses are also expected to occur in groundwater ecosystems. Their detection and characterization remain challenging due to methodological constraints and limited reference data, as most studies to date have targeted contamination[69,78] events rather than ecological processes. Nonetheless, their potential influence on microbial dynamics and biogeochemical carbon cycles should not be overlooked, as demonstrated in other environments, like the ocean[10]. Future RNA viromics efforts are essential to characterize this component of the virosphere and integrate it with our DNA-based framework.

Despite these advances, there are challenges and caveats to this kind of work. Towards host prediction, we used in silico analyses to identify potential hosts[21,30,57], which in the future could be complemented with experimental approaches, such as single-cell[79], Hi-C[80], epic-PCR[81], and recent integrated single-molecule DNA fluorescence in situ hybridization (FISH)-highly multiplexed ribosomal (r)RNA-FISH[82] approaches are needed to validate predicted virus-host interactions. Additionally, the lack of cultured representatives for CPR/DPANN and their hosts presents a major challenge in characterizing infection dynamics and evolutionary relationships. Cultivating these organisms[83,84] and their associated viruses will be essential for resolving key mechanistic questions, including how viruses infect ultrasmall prokaryotes, evade host defenses, and exploit host machinery as inferred through time-resolved multi-omics virocell experiments in other systems[6,7]. Finally, while not yet done in any ecosystem, future research will strive where it is able to more comprehensively and simultaneously evaluate the ecological diversity and ecosystem roles of all biological entities, including RNA viruses[10], plasmid[85], and mobile genetic elements[86]. Only in composite will be positioned to then place these into mechanistic modeling frameworks that provide and test hypotheses about viral or other entity's roles in community metabolisms that dictate ecosystem functioning and stability[87]. This current work provides a foundation for investigating groundwater-specific virus-host interactions that will ultimately help refine our ability to measure and understand viral contributions to microbial evolution, adaptation, and biogeochemical cycling in the groundwater microbiome.

## Methods

### Sample collection, DNA extraction, and metagenomic sequencing

Groundwater metagenomes were generated from samples collected at the Hainich CZE in Thuringia, Germany, including 31 samples from January 2019 (previously described)[57], and 34 new samples from November 2022 (Supplementary Data 1).

Sample handling, phenol:chloroform-based DNA extraction, and library preparation approaches were consistent across all years of sampling. The only difference between the 2019 and 2022 datasets lies in the sequencing platform used (see details in Supplementary Data 1). Groundwater samples (up to 100 L, in triplicate) were collected from seven wells (H14, H32, H41, H43, H51, H52 and H53) distributed along a ~ 6 km hillslope transect. The samples were sequentially filtered through 0.2 μm and 0.1 μm filters (Omnipore Hydrophilic PTFE membrane, Merck Chemicals GmbH). All the filter fractions were immediately frozen and stored under − 80 °C. DNA was extracted from microbial cell pellets using a phenol:chloroform protocol[88] adapted to preserve high-molecular-weight nucleic acids. Briefly, pellets were incubated in 2.6 ml SET buffer (Sucrose, Tris, EDTA) with 300 μl 10% SDS, 30 μl 100 mM PMSF in 2-propanol, and 1.5 g 0.3 mm silica carbide beads. The mixture was vortexed at maximum speed for 15 min and incubated for 60 min at 60 °C with shaking. After the addition of 3 ml phenol:chloroform:isoamyl alcohol (25:24:1, v:v:v), the mixture was vortexed for 10 min and incubated again for 60 min at 60 °C. Two subsequent extractions with PCI and a final extraction with

chloroform:isoamyl alcohol (24:1, v:v) were performed. DNA was precipitated overnight at −20 °C using glycogen (2 μl of 20 μg/μl), 1 ml 7.5 M ammonium acetate, and 8 ml 100% ethanol. Pellets were recovered by centrifugation (3200 × g, 30 min, 4 °C), washed twice with 80% ethanol, dried at room temperature, and resuspended in 50 μl TE buffer (10 mM Tris, 1 mM EDTA, pH 8).

Metagenomic libraries were prepared using the NEBNext Ultra II FS DNA Library Prep Kit (New England Biolabs), following the manufacturer's protocol. The only variation between years was the sequencing platform: metagenomes from 2019 were sequenced on an Illumina NextSeq 500 system (paired-end 2 × 150 bp), and those from 2022 were sequenced on an Illumina NovaSeq 6000 system (paired-end 2 × 150 bp). The average insert size was approximately 300 bp. In total, 65 metagenomes were generated, comprising 8,737,844,712 reads (range = 134,996,654–100,861,616; Q1 = 112,553,891; Q3 = 153,430,690). Sampling wells were categorized into two distinct groups based on location and oxygen availability[30,34,57]. Wells H41 and H51 are oxic and situated in flatter regions connected to the main aquifer, which is composed of limestone-rich Upper Muschelkalk. Wells H14 and H32, also oxic, are located in uphill zones. In contrast, wells H43, H52, and H53 are found in downhill areas and are characterized by suboxic or anoxic conditions (Fig. 1).

A previous study demonstrated that long-read sequencing substantially enhances the recovery of vOTUs[35]. To complement the short-read metagenomic data, we incorporated six groundwater samples (corresponding to the 0.2 μm filter fraction) for long-read sequencing (Oxford Nanopore), selecting those with the highest DNA concentrations. DNA extraction for long-read sequencing was performed using the DNeasy PowerWater Kit (HB-2267; Qiagen, Germany) following the manufacturer's protocol, with a minor modification to the cell lysis step: we applied a gentler method, vortexing at maximum speed for 5 min. Initial attempts to extract high-quality DNA from these samples using a phenol:chloroform protocol yielded poor results due to the presence of environmental impurities, especially when processing large volumes (hundreds of liters). In contrast, the PowerWater protocol provided cleaner, higher DNA concentration, more suitable for Nanopore sequencing. No additional size-selection or high-molecular-weight enrichment steps were applied. The extracted DNA was directly used for long-read library preparation and sequencing. Nanopore libraries were prepared using approximately 500 ng of metagenomic DNA and the ligation kit (SQK-LSK110, ONT). The detailed numbers of reads and bases can be seen in Supplementary Data 1.

The groundwater hydrochemical composition was described elsewhere[34,89]. Briefly, we measured a range of environmental and physicochemical parameters, including water table depth, temperature, pH, electrical conductivity standardized to 25 °C (EC$_{25}$), dissolved oxygen, redox potential (ORP), as well as buffering capacities for both acidic and basic conditions. Additional analyses encompassed total inorganic carbon (TIC) and concentrations of key cations and anions. For elemental analysis, concentrations of calcium, potassium, magnesium, sodium, and sulfur were determined using inductively coupled plasma mass spectrometry (ICP-MS; Agilent 8900 Triple Quadrupole, Germany). Chloride levels were measured separately through ion chromatography (Dionex IC20, Thermo Fisher Scientific, USA). Dissolved organic carbon (DOC) was quantified based on non-purgeable organic carbon content, employing a vario TOC cube analyzer (Elementar Analysensysteme, Germany), with a minimum detection limit of 0.5 mg L$^{-1}$. Methodologies followed established protocols described by Lehmann and Totsche[89] and Kohlhepp et al.[34].

### Metagenomic processing, identification, and abundance quantification of MAGs

Metagenome-assembled genomes (MAGs) were generated as previously described[57]. Briefly, metagenomic sequencing was quality-filtered via the bbduk script (BBMap version 39.01). Next, SPAdes

(version 3.15.2) was used to assemble the contigs in−meta mode with default parameters[90]. Contigs longer than 1000 bp were binned using five different binning tools with default settings: (*i*) binsanity (version 0.2.7)[91], using a minimum scaffold size of 3 kbp, (*ii*) abawaca (version 1.0.0) using 5 kbp, and minimum scaffold size, (*iii*) maxbin2 (version 2.2.6)[92] using 40 and 107 gene markers, (*iv*) CONCOCT (version 1.0.0) and (*v*) metabat2 (version 2.12.1)[93]. Binning results were refined using Metawrap refinement module (version 1.3.2) with filters -c 50 -x 10[94], followed by selection of the best representative MAGs using dRep (version 3.4.0)[95]. To identify Candidate Phyla Radiation (CPR) genomes, we used the *anvi-script-gen-CPR-classifier* from Anvi'o v6.1[96], which applies a random forest classifier to predict the probability of belonging to CPR. From the initial set of 1778 dereplicated metagenome-assembled genomes (MAGs), of high and medium quality, we applied standard genome quality criteria to retain only those of medium or high quality for downstream analyses, including: (*i*) a quality score (QS = completeness − 5 × contamination), with bins scoring QS < 50 being removed[97]; (*ii*) retention of high-quality bins with completeness ≥ 90% and contamination ≤5%; and (*iii*) retention of medium-quality bins with completeness >50% and contamination ≤10%[98]. Following this refinement process, we retained 1275 MAGs (high- and medium-quality) for the downstream analyses, including subsequent in silico virus-host predictions. A genome-quality assessment was carried out using the CheckM workflow (version 1.1.3)[99]. Lastly, the taxonomic annotations of the MAGs selected for analysis were carried out with GTDB-Tk (version 1.5.1)[100] using GTDB (release 202) as a reference database.

For long-read processing, the samples were then individually sequenced on MinION flow cells (FLO-MIN106 with an R9.4.1 pore) for 72 h. Basecalling was performed using Guppy (Version 6.0.1) with a super-accurate model (dna_r9.4.1_450bps_sup.cfg). Next, hybrid assemblies were generated using Spades Hybrid with the long-read fastq files. A total of 17 hybrid assemblies were generated from samples belonging to the 0.2 μm filter fraction (see Supplementary Fig. 1). To create a MAG abundance table, we calculated trimmed mean abundances using CoverM[101] (-m trimmed_mean) with the following parameters: −min-read-percent-identity 0.95, −min-read-aligned-percent 0.75 −min-covered-fraction 0.25) (https://github.com/wwood/CoverM). Trimmed mean abundances were normalized by the number of quality-controlled reads per sample per terabase. Approximately 25.9% of metagenomic reads mapped to the MAGs, indicating a substantial proportion of the microbial community is represented by the reconstructed genomes. The accumulation curves for these MAGs demonstrated overall saturation across the groundwater metagenomes, suggesting that most of the microbial diversity captured by MAGs has been adequately sampled (Supplementary Fig. 14e).

### Virus identification, filtering, virus clustering, and quality assessment

Viromics pipeline is illustrated in Supplementary Fig 1. For virus identification, we used assembled contigs generated from metaSPAdes as described above. To enhance viral genome recovery, we additionally assembled the metagenomic reads using MEGAHIT v1.1.3 (default)[102], as different assemblers can yield complementary viral contigs due to distinct assembly heuristics. MEGAHIT has been benchmarked as effective in recovering low-abundance or fragmented viral genomes[103]. Further, we utilized VirSorter2 (version 2.2.3)[104], DeepVirFinder (version 1.0)[105], VIBRANT (version 1.2.1)[106], and geNomad (version 1.5.1)[44] to detect viruses within assembled contigs from each sample. To ensure the confidence of the identified viruses, we applied the following filters. First, we followed the recommended thresholds for each tool: (*i*) for VirSorter2 we applied standard operational procedure (protocols.io/view/viral-sequence-identification-sop-with-virsorter2-5qpvoy-qebg4o/v3). (*ii*) for DeepVirFinder, the score was ≥0.9, and the *e*-value was ≤0.05. (*iii*) VIBRANT, default settings, and (*iv*) geNomad, default

settings. Given its conservative nature, CheckV serves as a valuable tool for identifying false positives[107] and distinguishing viral from host genes. This aligns with prior research that has explored the necessity of additional filtration steps aimed at removing non-viral sequences[73]. Hence, we applied a second filter based on CheckV, as recommended in previous studies[104,108], and manually spot-checked contigs from DeepVirFinder, VIBRANT, and geNomad. Specifically, we retain virus contigs meeting one or more of the following: (*i*) >0 viral genes, (*ii*) virus gene = 0 and host gene = 0, (*iii*) virus contigs length ≥1 kb, and (*iv*) virus contigs with genes ≥75% unknown genes[108]. Lastly, we trimmed host-associated regions using the CheckV "end-to-end" function and default settings (see Fig. 1 for detailed workflow and contig numbers). To ensure high confidence, we also employed manual spot checks (Supplementary Fig. 2).

The resulting 4,708,626 virus contigs were clustered into a viral operational taxonomic unit (vOTU) using MMSeq2[109], applying 95% average nucleotide identity (ANI) across ≥80% of the shorter alignment sequence[36]. This yielded 2,412,499 vOTUs ≥1 kb, of which 257,814 were ≥5 kb, and 82,807 were ≥10 kb in length. To remove likely artifacts such as degraded cellular genomic islands, or mobile elements, we applied a curation filter[41] to vOTUs ≥100 kb. In this filter, we exclude any contigs that contained genes annotated as transposons, lipopolysaccharide genes (glycosyltransferase, nucleotidyl transferase, carbohydrate kinases, nucleotide sugar epimerase), endonuclease, integrase, or plasmid stability since such genes are enriched in the genomic island (reviews see Dobrindt et al.[110]. and Bertelli et al.[111].). This removed 562 vOTUs >100 kb, resulting in a final conservative dataset of 257,252 vOTUs ≥5 kb, including 82,245 vOTUs ≥10 kb. We further used CheckV[107] to assess the quality and completeness of vOTUs, and VIBRANT[106] to predict virus lifestyle (virulence vs temperate, both using default settings) (Supplementary Data 2).

### Calculating virus abundance and ecological statistics

To create a virus abundance table, we calculated trimmed mean abundances (-m trimmed_mean), using CoverM[101] (--min-read-percent-identity 0.95 --min-read-aligned-percent 0.75 --min-covered-fraction 0.7) (https://github.com/wwood/CoverM). Next, we normalized the trimmed mean abundance table by dividing the abundances by the number of quality-controlled reads per sample per 1 Tb (Supplementary Data 4). We further assessed species accumulation curves using the vegan specaccum() function in R, providing insights into the accumulation of distinct viral species across samples.

We next examined the distribution of each vOTU across the groundwater samples. We considered viruses to be present if their normalized vOTUs relative abundance exceeded 0 in at least one sample. Using a similar approach, we further categorized vOTUs as unique if found in one sample, and shared if found in multiple samples[9]. Additionally, we classified vOTUs as core (present in >50% of the wells), common (21–50%), and rare (1–20%). We visualized vOTU relationships across the well using UpSet plot[112] (UpSetR; Supplementary Figs. 6, S9), and stacked bar plots (ggplot2[113]; Supplementary Fig. 9b, c).

To explore finer-scale similarity among viral genotypes and account for potential convergent evolution, we implemented a re-clustering approach. Specifically, we re-clustered the virus contigs at thresholds of ≥99.5%, ≥99.8%, and ≥100% of average nucleotide identity. We then visualized the results using the UpSetR package in R (Supplementary Fig. 6). Ecological analyses were performed in R using the "vegan" package. First, we log-transformed the normalized relative abundance table from the previous section and generated Bray-Curtis dissimilarity (β-diversity) matrices as input for Principal Coordinate Analysis (PCoA, cmdscale() function) (Supplementary Fig. 10) and Non-metric MultiDimensional Scaling analysis (NMDS, metaMDS() function) (Fig. 2a, and Supplementary Fig. 10). For PCoA, the emerging groups were statistically verified using PERMANOVA tests with

adonis() function. For NMDS, statistical significance was determined using the anosim() function and multiple response permutation procedure (mrpp() function, distance = euclidean, permutations = 9999). We calculated α-diversity metrics, including Shannon's $H$, Richness, Pielou's $J$, Simpson, Inverse-Simpson, and evenness, using the "vegan" R package[114]. In the main text, we referred to Shannon's $H$ as a macrodiversity (inter-population or species diversity) (Supplementary Data 6).

Pearson's correlation and Mantel test were performed to assess the pairwise relationships between the environmental factor data (18 factors), viruses (Shannon's $H$, macro-, and microdiversity), and their respective hosts (Shannon's $H$, and richness) (Supplementary Fig. 17). The physicochemical variables used (Supplementary Data 7) included: water level (MAMSL), water temperature (°C), specific electrical conductivity (μS/cm), pH, redox potential (mV), dissolved oxygen content (μmol/L), acid neutralizing capacity (Alkalinity) (meq/L), base neutralizing capacity (acidity) (meq/L), total inorganic carbon (μmol/L), chloride (μmol/L), calcium (μmol/L), potassium (μmol/L), magnesium (μmol/L), sodium (μmol/L), sulfur (μmol/L), nitrate (μmol/L), ammonium (μmol/L), sulfate (μmol/L)[56]. All analyses and data visualization were conducted using the linkET package (https://github.com/Hy4m/linkET) in R.

Virus microdiversity was calculated as previously described[9], using MetaPop v0.0.48[115] (--min_cov 70). Briefly, viral populations meeting the criteria of an average read depth of ≥10x across 70% of their contig in at least one sample were analyzed for microdiversity. Single-nucleotide variants (SNVs) were called from BAM files with reads mapping at ≥ 95% nucleotide identity, using a quality threshold of >30. Nucleotide diversity (π) per genome was calculated following downsampling to 10x coverage per locus. The final microdiversity value (average π) for each sample was determined by averaging π values from 100 randomly selected viral populations across 1000 subsamplings (Avg. π).

### Virus taxonomic classification

There is a wide range of tools available for virus taxonomy. We applied two complementary approaches to classify all vOTUs ≥10 kb. First, used the recent virus taxonomy tool vConTACT3[46] v.3.1.4 (https://bitbucket.org/MAVERICLab/vcontact3/src/master/) for all the vOTUs ≥ 10 kb. The vConTACT3 classifies viruses into taxa based on the gene-network approach and is aligned with the International Committee on Taxonomy of Viruses (ICTV). vConTACT3 uses NCBI Virus RefSeq release 230 and incorporates host information from the Virus-Host DB (using GenBank release 257). We assigned taxonomy to the vOTUs using vConTACT3 with the default setting. The taxonomy assignment is summarized in Supplementary Fig. 4, with full results provided in Supplementary Data 3.

Second, we also used geNomad, which classifies viruses into taxa based on the International Committee on Taxonomy of Viruses (ICTV) taxonomy (version 2022) and its extensive collection of over 85,000 markers, each distinctly linked to various virus taxa[44]. We used geNomad default setting ("virus_score" ≥0.7) to assign the taxonomy of the vOTUs. The complete results can be found in Supplementary Data 3.

### Viral clustering and database comparison

To compare the similarity of Hainich groundwater viruses with other publicly available databases, we downloaded the following databases: IMG/VR v4.1[37] (5 M vOTUs), Gut Virome Database (GVD, 33,242 vOTUs)[47], Global Soil Virome database (GSV, 80,750 vOTUs)[48], Global Oceans Viromes 2.0 database (GOV2, 195,728 vOTUs)[9], Viral RefSeq v.222[49] (18,719 vOTUs), Fennoscandian Groundwater Virome (FGV, 4,051 vOTUs)[39], Rumen Virome Database (RVD, 397,180 vOTUs)[50] and Groundwater Virome Catalogue (GWVC, 280,420 vOTUs)[40]. To ensure an equal comparison, we further filtered vOTUs ≥ 10 kb in length. This process yielded refined datasets: GVD (15,330 vOTUs), GSV (80,750 vOTUs), GOV2 (195,728 vOTUs), Viral RefSeq v.222[49] (6,574 vOTUs),

FGV (1,407 vOTUs), RVD (193,350 vOTUs), and GWVC (107,610 vOTUs). For IMG/VR, we applied the following filters: (i) vOTUs ≥ 10 kb in length; (ii) only freshwater-origin samples (i.e., lake, river, lentic, groundwater, creek, pond, lotic, and reservoir); (iii) high-quality genome only; and (iv) exclusion of RNA viruses and sequences lacking taxonomic assignment. This resulted in 558,408 IMG/VR entries from freshwater environments.

We assessed similarities at both the vOTU- and protein cluster-level against GOV2, GSV, GVD, virus RefSeq, RVD, GWVC, and IMG/VR (groundwater subset). For vOTU-level comparisons, we used three approaches. First, we performed a BLASTn search, as previously described[9]. The vOTUs showing nucleotide alignment with ≥ 95% nucleotide identity and ≥ 50% of an alignment length were considered present in other databases (Supplementary Fig. 5). We predicted protein-coding sequences from all viral contigs in this study and the reference databases using Prodigal (v2.6.1)[116]. To standardize the datasets, we filtered out short sequences and retained only proteins ≥100 amino acids in length. This resulted in 1,703,721 proteins from our dataset (out of an initial 2,592,284), compared to 3,618,709 from GOV2, 2,089,631 from GSV, 446,574 from GVD, 95,888 from IMG/VR groundwater, 450,852 from viral RefSeq, 25,811 from FGV, 5,023,585 from RVD, and 2,456,943 from GWVC. We then performed protein-level clustering using MMSeq2 with the following parameters: --min-seq-id 0.3 -c 0.6 -s 7.5. We also ran MMSeq2[41] for vOTU-level comparisons using stricter thresholds (--min-seq-id 0.95 -c 0.5; --min-seq-id 0.95 -c 0.8) to assess species-level overlap between our dataset and IMG/VR as well as GOV2 (see Supplementary Fig. 5).

### Host predictions and lineage-specific virus/host abundance ratios (VHRs)

We used a recently developed virus host prediction tool, iPHop (version 1.3.2)[61]. iPHop integrated BLAST and CRISPR-spacer similarity with oligonucleotide frequency (ONF)-based distance/dissimilarity measures, $k$-mer frequencies, a Gaussian Model (GM), and a machine-learning approach on protein clusters to assign hosts.

For our analysis, we used two approaches: (i) a default databases (iPHoP_db_Sept21_rw), which also includes host genomes extracted from GTDB r202, IMG published genomes as of Sept. 2021, and GEMv (Earth's Microbiomes catalog); and (ii) 1275 metagenome-assembled genomes (MAGs) obtained from the same site, Hainich (Supplementary Fig. 14 and Supplementary Data 8: second tab "MAGs"). For the default, we used the "predict" command with the "Sept_2021_pub_rw" database. In the second approach, we followed the recommended workflow (https://bitbucket.org/srouxjgi/iphop/src/main/), and symbolically linked our 1,275 MAGs to the original database using "add_to_db". Given that longer virome fragments (≥10 kb) improve host prediction accuracy by increasing recall rates and reducing false discovery rates (FDRs)[61], we applied this threshold to our dataset. We combined cellular genomic data and predicted hosts for our 82,756 vOTUs (≥10 kb).

To this end, we integrate the results from both approaches and considered only the top hits of virus-host links for further analysis. VHRs were calculated following Emerson et al.[51]. Specifically, we used the average per-base coverage depth of each vOTU and its predicted host MAG, derived from read mapping and normalized by sequencing depth, to compute virus-host ratios. The lineage-specific virus/host abundance values can be seen in Supplementary Data 8, 9.

We note that MAG dereplication and vOTU clustering may obscure fine-scale CRISPR variation. However, this approach aligns with iPHoP's best practice and benchmarking at the genus level and its helps reduce redundancy and noise in downstream analyses[61].

### CRISPR-Cas spacer identification and screening against viruses

To identify the CRISPR-Cas and spacers (≥3 direct repeats) from these MAGs, we used MinCED v0.4.2 (https://github.com/ctSkennerton/

minced). We then matched the spacers against vOTUs (82,245; ≥10 kb) allowing ≤1 mismatch over ≥95% of the spacer length using BLASTn (-word_size 7 -task "blastn-short" -evalue 1e-5). Allowing up to two mismatches in CRISPR-Cas spacers has been reported to represent as an optimal tradeoff between capturing real virus–host interactions and minimizing false predictions[117]. The spacers were also matched against IMG/VR spacer databases (Virus DNA DB ver.4 2022-09-15, Viral Spacers Public 2025-01-22, Metagenome Spacers Public 2025-01-22).

### Identification of virus-encoded auxiliary metabolic genes (AMGs)

To identify virus-encoded putative auxiliary metabolic genes (AMGs), we used DRAM-v[72] and followed the recommendation to analyze only ≥10 kb in length. DRAM-v annotated vOTUs using a variety of databases, including Pfram, KEGG, UniProt, CAZyme, MEROPS (the peptide database), VOGDB (Virus Orthologous Groups Database), and NCBI viral RefSeq to annotate virus genomes. Before running DRAM-v, we used VirSorter2 (--prep-for-drama) to generate the required input files. We then applied several recommended filters to identify putative AMG, including: (*i*) retained those with AMG category 1-3, and (*ii*) those with AMG flag M. Conversely, sequences with AMG category 4 or flags V, A, P, or B were considered non-AMGs. To increase the confidence of putative AMGs (after DRAM-v curation), we refined the analysis by excluding viral genes commonly misclassified as metabolic genes. For example, we exclude genes encoding enzymes such as dUTP pyrophosphatase[118], DNA (cytosine-5-)-methyltransferase[119], glycosyl transferase[120], ribonucleoside-triphosphate reductase[121], and ribosomal subunit genes. Additionally, we also annotated the microbial MAGs (1275) using DRAM[72]. We further clustered the putative AMGs using MMSeq2 (--cluster-mode 2 --cov-mode 1 -c 0.6 -s 7.5 --kmer-per-seq 20) as previously described[71].

To delve deeper into the influence of putative AMGs on the metabolic pathways of groundwater microbial communities, we used KEGG mapper construct[122]. Initially, genes associated with KEGG Orthology (KOs) were mapped onto metabolic pathways using KEGG mapper reconstruction. This step allowed us to record two key parameters: the number of unique KEGG identities (no_kegg_ID) involved in module pathways, and an assessment of module completeness (module_comp). Concurrently, a similar analysis was conducted for putative AMGs-identified KEGGs, allowing for a comprehensive understanding of their role within these pathways. The results were then visualized using the geom_tile() functions from the ggplot2 package. Furthermore, we considered viruses to target specific metabolic modules if a putative AMG was present within the corresponding pathways. This integrated approach provided valuable insights into the intricate interplay between microbial and viral components within groundwater microbial communities and their metabolic activities.

### Metatranscriptome analysis

The metatranscriptomics dataset (initially consisting of six wells with a total of 18 samples) and the analysis were described elsewhere[15]. Briefly, groundwater subsampled were collected in August and November 2015, for well H32, H41, H42, H43, H51, and H52. We excluded the metatranscriptomic data from well H42, because this well was not sampled for the metagenomic analysis. After excluding data from well H42, we used a total of 17 metatranscriptomic datasets. Briefly, groundwater subsamples for metatranscriptomic analysis were collected in August and November 2015, for wells H32, H41, H42, H43, H51, and H52. The QAQC-filtered reads were mapped to MAGs using Bowtie2 v.2.3.5 in sensitive mode[123], and the total number of *rpoB* transcripts from each metatranscriptomic library was determined. Transcript abundances were quantified using Kallisto package (version 0.48.0)[124] and normalized using scaling-factor based on the total number of *rpoB* reads from the QAQC filtered metatranscriptomic

reads. Additionally, for a viral population to be considered detected in a metatranscriptome, there must be an average coverage value for at least one gene for every ten kilobases (kb) of viral genomic sequence, as recommended previously[51]. These methods estimate active viral community composition, but their accuracy remains uncertain due to the lack of standardized approaches or biological benchmarks for assessing viral "activity" in bulk metatranscriptome data. We further visualized these active viruses using a heatmap generated with Flaski (https://github.com/mpg-age-bioinformatics/flaski)[125], applying log10 scaling to the sum of relative transcript expression values (for viruses in Supplementary Data 4). The scaling factor for *rpoB* can be seen in Supplementary Data 4.

We also analyzed metatranscriptomic data for MAGs as described above. Genes were included if they were expressed in at least 10% of the samples. We also assess relative expression without applying that cutoff as previously described[15]. In addition to scaling factor normalization, we further normalized relative gene expression using the average genome size of MAGs within each phylum. The average genome size was determined by estimating genome lengths from our MAGs (observed genome length/completeness) and integrating reference genomes from IMG/G (last accessed: 24 February 2025). This dataset includes genomes from prokaryotic isolates (115,873 bacterial and 1379 archaeal genomes) and MAGs (14,481 bacterial and 1002 archaeal genomes).

### Co-occurrence network analysis for MAGs

To identify a possible host for Candidate phyla radiation (CPR), we used a previously described approach[57]. We used the normalized average genome coverages table encompassing 1275 MAGs across all metagenomes. We then used this abundance matrix to calculate the proportionality of the coverage profiles in R package propR v5.0.2[126]. We applied the following thresholds: (*i*) ρ a threshold of ≥0.95 to capture the strongest co-occurrences patterns, (*ii*) only associations involving CPR and DPANN were considered, and (*iii*) only positive correlations. The resulting network was visualized using Cytoscape v3.10.1[127].

### Reporting summary

Further information on research design is available in the Nature Portfolio Reporting Summary linked to this article.

## Data availability

Source data are provided with this paper. The data used for this study have been deposited in the European Nucleotide Archive (ENA). Raw metagenomic sequencing reads from 2019 were deposited under ENA project accession PRJEB36505 (https://www.ebi.ac.uk/ena/browser/view/PRJEB36523), while 2022 sequencing reads were deposited under NCBI project accession PRJNA1236243. Processed data are available through Zenodo, including viral populations (≥5 kb), Dereplicated MAGs, putative AMG genes, putative AMG cluster genes, and identified CRIPSR-Cas spacers (https://doi.org/10.5281/zenodo.17897233). Source data are provided with this paper.

## Code availability

Workflows, codes and data related to this manuscript are available at https://github.com/AAdjieP/Groundwater_virome/ or https://doi.org/10.5281/zenodo.17898531.

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

## Acknowledgements

Akbar Adjie Pratama acknowledges financial support provided by the Deutsche Forschungsgemeinschaft (DFG, German Research Foundation) through Germany's Excellence Strategy (EXC 2051, Project-ID 390713860), awarded to K.K. and M.B.S. Additional support was received from the Collaborative Research Centre AquaDiva (CRC 1076, Project-ID 218627073) awarded to K.K., provided by the Deutsche Forschungsgemeinschaft (DFG, German Research Foundation). Part of this work was enabled by funding provided to M.B.S. by the U.S. Department of Energy, award #DE-SC0023307. The authors would like to thank Heiko Minkmar, Falko Gutmann, René Maskos, and Stefan Riedel for groundwater sampling and on-site measurements/sample preparation. The authors would also like to thank Olivier Zablocki for his input on the manuscript. Thank you to Benjamin Bolduc for his help in interpreting the virus taxonomic analysis.

## Author contributions

A.A.P., K.K., and M.B.S. created the study design. O.P.C. and A.A.P. collected all datasets. O.P.C. and A.A.P. performed the data analysis and visualization. A.A.P., O.P.C., M.B.S., and K.K. contributed to the scientific discussion and wrote the manuscript. All authors read and approved the final manuscript.

## Funding

## Competing interests

The authors declare no competing interests.
