## [Transparent Peer Review file · Nature Communications]

Diversity and ecological roles of hidden viral players in groundwater microbiomes

Corresponding Author: Professor Kirsten Küsel

Version 0:

Reviewer comments:

Reviewer #1

(Remarks to the Author)

This manuscript presents an analysis (and new metagenomic data) focused on groundwater viruses. The system is interesting and the authors attempt to compare it with other systems to put it in context. There is a large focus on tiny prokaryotes (CPR and DPANN) that are prevalent in groundwater. These are all topics of timely interest to the community. However there is a lot of speculation and spinning threads based on limited evidence, which should be toned down significantly.

Major findings:

- Identified ~257k vOTUs (dereplicated viral contigs >5kb) in groundwater, far more than previous groundwater datasets; mostly Caudoviricetes though 30% unassignable to phylum, with a high degree of novelty (i.e. never before detected in other datasets; but see below for comments on the approach)
- Mapped some (a small fraction, ~11%) of vOTUs to predicted hosts; some mapped to CPR and DPANN organisms
- Estimated virus:host ratios via read mapping to MAGs and vOTUs, found variation across taxa (but see comments below)
- Identified AMGs in these vOTUs related to biogeochemical (C,N,S) cycling
- Found varying degrees of viral overlap between sampling wells, reflecting aquifer hydrology and environmental conditions (esp oxygen)

Strengths:

- large dataset of groundwater metagenomes (a poorly characterized habitat), specifically analyzed for viruses. Includes 0.5Tb of existing MG data plus 0.7Tb new MG data, plus 22Gb existing metatranscriptome data
- Good motivation w.r.t. biogeochemistry, prevalence of tiny cells, extremely oligotrophic environment dominated by chemolithoautotrophy
- discovery of new viruses predicted to infect taxa such as CPR, which are poorly characterized
- Introduction is generally well written and motivates the study

Weaknesses:

- very weak evidence to support some threads in the manuscript (novelty, virus decoys, tripartite interactions, influence of AMGs on the system)
- database comparison to assess novelty: The authors found no vOTU overlap between their groundwater dataset and several other viral datasets from various environments. it is impossible to assess whether groundwater viruses are especially novel without some context. What if you used the same procedure to compare marine and freshwater? or two different freshwater datasets? If there is no overlap in these other comparisons either, then the groundwater novelty is unremarkable (by this approach).
- emphasis on the results from host predictions, but only 11% of the dataset could be matched to a predicted host. Therefore it is misleading to extrapolate from the 11% to the whole system. The section about virus decoys and alternative genetic codes (lines 280-303) is supported by very little evidence in the current paper and is extremely speculative. Alternative explanations are not explored (e.g. very low burst size in CPR, lots of unidentified CPR viruses in the 89% of vOTUs that could not be matched to a host)

-no evidence of tripartite interactions. This whole section and fig 3d-e should be removed. The authors attempt to identify CPR/DPANN host associations via co-occurrence network analysis. However the co-occurring taxa need not be hosts for the CPR/DPANN -- more likely, they are simply co-responding to similar environmental drivers. E.g. chloroflexi and nitrifiers also like dark, low nutrient environments. There is no evidence they are physically associated with CPR/DPANN. Therefore, there is no evidence that viruses are somehow involved in a tripartite interaction here. This section is pure speculation.

-AMGs of CPR/DPANN section has a lot of redundancies and speculation, based on limited evidence. line 415 claims "increased presence of AMGs" but relative to what? It is not demonstrated that CPR or groundwater have more AMGs. Likewise "prevalence of AMGs related to carbon and energy" and the comparison to deep sea where there is "an enrichment of AMGs involved in central carbon metabolism" is not supported. Need statistics to show enrichment (and enriched compared to what?).

-Lots of incomplete sentences and typos in text and figure legends, esp in Methods

-Too long and repetitive in many places

Specific comments:

-lines 31-33: combine 1st very short paragraph with subsequent. Suggest not focusing the first sentence of the m.s. on "phages" if you also discuss viruses of Archaea.

-Intro: remove carrot symbols for exponents

-line 52: "rates comparable to those in sunlit oceanic water" -- do you mean comparable to rates of chemolithoautotrophy in the surface (which are probably very low), or rates of photoautotrophic C fixation (which are much higher)?

-line 66: "viral ecogenomic" ?

-line 71: what do you mean by "selectively infecting"? Of course the viruses have to be specific for the taxa present

-line 72 and elsewhere in text and figure legends: remove "pristine". Ill defined and subjective description (there are probably PFAS in the groundwater...)

-line 70-73: I'm not sure what the difference is between the two hypotheses/proposals ("We hypothesize" and "we further propose").

-Fig 1 legend part b: "with the numbers representing sampling replicates" -- which numbers are you referring to?

-Methods - Sampling, lines 472-3: "A total of 50 and 100L..." it is not clear how this was sampled. The legend for fig 1 says sequential filtering, which implies that the same volume would be on both filters. was more volume collected on 0.1? Were there triplicate 0.1 and 0.2 filters for each well?

-How was the 0.1um filter used vs the 0.2? Did you directly compare the results from the two size fractions? It would be interesting to know if the 0.1 captured different viruses, and your reasoning for sampling both sizes. (there is only a brief mention in line 190 and ordination plots).

-Fig S1 workflow: in "Filtering 3" step, it says "specifically those >100kb". but the input and output from that step are >5kb...

-Methods: there are many incomplete sentences throughout, and other grammatical errors, and sentences that say "I did x..." -- Were the methods carefully read and edited by all authors?

-citations: there is some sloppy referencing in the text. e.g. line 112-3: are refs 13 and 19 for the datasets? unclear how they relate. Line 116-118: are there other refs you can cite besides permafrost/arctic -- e.g. marine or freshwater? refs 49-50 are gut, unrelated (they seem to be supporting the definitions of core, common, and rare -- but these are intuitive and don't need citations). ref 52 (OMZ) seems unnecessary and potentially misleading, unless you are also going to mention other oxic and anoxic systems (not just the ones published by your group). Ref 53 also seems unnecessary, unless you are going to cite other examples of environmental factors influencing viral communities. Line 271-3: again seems like a cherry picked result and does not contribute to the discussion at hand; please remove refs 34-35 here or include a broader discussion of VHRs across systems. Ref 74 is unnecessary, random, and out of place (line 409). ref 90 (line 517) is unnecessary (normalization is standard). Ref 112 is unnecessary.

-Fig S6: is gray active / black inactive? need legend

-line 118-120: host predictions come later, so perhaps move this sentence

-line 126-130: what's the difference between the Global Ocean Virus db and GOV - c.f. the 0.4-fold and 2.8-fold statistics

-line 134-136: was the initial comparison with IMG-VR limited to groundwater? specify here; because later you expand to IMG-VR-freshwater

-line 146-152: remove this paragraph; all you need is a concluding sentence in the previous paragraph summarizing the findings. Ref 47 is not relevant here, and well-specific patterns are discussed in the following section.

-line 162-4: incomplete sentence

-line 170-173: remove host prediction sentence, because that comes in a later section

-line 215-217: remove this sentence; the data don't actually show this, and the sentence is unnecessary.

-fig S14b: it's really hard to distinguish the C/O/F patterns. Why not show a tree instead?

-section starting line 219: earlier you found oxygen to be important; can you tie that in here? how does oxygen relate to these parameters? Concluding sentence is weak, can you revise (or remove)?

-fig 3a: unclear what is being plotted here. Is it the rel abund of these prokaryotes, or the rel abund of predicted hosts? i.e. was the overall abund of Proteobacteria in H14 75%? or, of the hosts predicted in H14, 75% were proteobacteria?

-host prediction section: The authors should emphasize that this whole discussion is based on just 11% of the vOTUs. So it is not fair to say that Proteobacteria are the most common host in this system. likewise for VHR discussion: please be explicit about the limitations. What fraction of the data are we actually able to quantify here? What fraction of the reads are represented by the host MAGs and vOTUs with host predictions? How might the conclusions change if the missing 89% of vOTUs infect CPR or DPANN or Proteobacteria (or aren't viruses at all)?

-line 249-250: SAR11 is Proteobacteria, so that doesn't contrast with your findings. An alternative explanation is that Proteobacteria and their phages are well represented in the dbs, and therefore they are more commonly identified.

-line 267: remove statement about "higher viral diversity" - the VHR does not specifically address viral diversity

-line 276-7: clarify that 18.42% is of host-matched vOTUs (only 11%)
-line 312-321: this describes host predictions, which are discussed in the previous section. Move these results to the previous section (where you already mention the same results, i.e. n=1771 CPR-linked vOTUs)
-line 371-372: please explain what the values mean ("45.6% relative abundance" - what is the numerator and denominator?)
-line 394-6: combine with previous paragraph and tone down the conclusion. I do not agree that "compelling evidence" has been presented for influencing metabolism, nutrient cycling, and "ecosystem adaptation" (whatever that is)
-line 398: AMGs in viruses of CPR and DPANN (AMGs are not in bacteria/archaea, they are in viruses)
-line 401-402: how does this relate to CPR metabolism? CPRs are fermentative anaerobes
-line 406: what is the point about chaperones and cell surface genes? how does that relate to metabolism?
-line 407-410: remove; does not fit with the narrative and is speculative.
-lines 424-8, 439-44 are redundant with other text. this section could be streamlined substantially.
-line 435: "19% of virus populations, with 9% observed" -- what do those values mean? 9% observed?
-line 437-8: "suggesting the ecological importance of AMGs are likely important" (grammar)
-line 449: "higher-than-expected" -- based on what? was there a stated hypothesis about the rarity of CPR/DPANN viruses? or just your hunch?
-line 517-519: 25% of reads mapping to MAGs does not seem very substantial. This is >75% in many aquatic systems. Be careful what you say about the microbial community based on just 25% of it.
-line 547: "transposons"

Reviewer #2

(Remarks to the Author)

The study instigates metagenomic and metatranscriptomic data to identify 257,252 viral operational taxonomic units (vOTUs), with 99% representing novel viruses, underscoring the vast uncharted viral diversity in groundwater ecosystems. This aligns with previous studies suggesting that subsurface environments harbor unique viral communities distinct from surface ecosystems. The high novelty of these viruses highlights the need for expanded viromic exploration in subsurface habitats to better understand their ecological roles.

Authors found that viruses in groundwater primarily infect Proteobacteria, Candidate Phyla Radiation (CPR) bacteria, and DPANN archaea—key microbial groups in subsurface ecosystems. A particularly intriguing observation is the low virus-to-host ratio in CPR bacteria, coupled with the presence of viral CRISPR spacers targeting multiple hosts. The authors propose a "virus decoy" mechanism, where CPR bacteria may act as viral sinks, absorbing infection pressure and thereby protecting more metabolically active bacterial hosts. This hypothesis, if validated, could reshape our understanding of viral predation dynamics in energy-limited ecosystems.

In addition, a major contribution of this study is the identification of 3,378 vOTUs encoding AMGs involved in carbon, nitrogen, and sulfur cycling. These AMGs suggest that viruses actively reprogram host metabolism, potentially enhancing nutrient turnover in groundwater. Notably, 31.5% of host metabolic modules were targeted by viruses, indicating a substantial viral influence on microbial metabolic networks. This finding supports the growing recognition of viruses as key players in biogeochemical cycling, extending beyond their traditional role as microbial predators.

Overall, this study advances our understanding of viral ecology in groundwater ecosystems, revealing unprecedented diversity, novel host interactions, and metabolic reprogramming capabilities. The proposed "virus decoy" mechanism and the widespread presence of AMGs underscore the need to integrate viruses into models of subsurface biogeochemical cycling.

However, there are several Limitations and Open Questions

1. Lack of Cultured Representatives

While the study identifies numerous novel viruses, the absence of isolated viral-host pairs limits mechanistic validation of the proposed interactions. Future studies should prioritize cultivation-based approaches to confirm host specificity and infection dynamics, particularly for CPR bacteria and DPANN archaea.

2. Limited Comparative Analysis with Other Groundwater Systems

The study focuses on seven wells in the Hainich CZE, which, while valuable, represents a relatively narrow sampling scope. Comparative studies incorporating data from other groundwater ecosystems (e.g., pristine vs. contaminated aquifers, deep vs. shallow groundwater) would strengthen the generalizability of the findings. Additionally, benchmarking against existing groundwater virome datasets could help discern whether the observed viral diversity and host interactions are site-specific or broadly representative.

3. Dynamic Viral-Host Relationships Require Further Investigation

The study provides a snapshot of viral diversity, but longitudinal sampling could reveal how viral communities shift in response to seasonal or anthropogenic perturbations. Although the authors employed iPHoP (a machine-learning-based host prediction tool integrating BLAST, CRISPR-spacer matching, oligonucleotide frequency, and protein clustering), host assignment remains probabilistic. Experimental validation (e.g., single-cell genomics or phage isolation) would enhance confidence in the predicted interactions, particularly for the proposed "virus decoy" mechanism.

4. Quantifying the Ecological Impact of AMGs

While AMGs suggest metabolic reprogramming, their actual contribution to biogeochemical fluxes remains unclear. Future studies could employ stable isotope probing (SIP) or metabolomics to assess whether viral AMGs directly influence carbon, nitrogen, or sulfur cycling rates. Additionally, transcriptomic and proteomic profiling of infected hosts could help determine the functional expression of these AMGs under in situ conditions.

Reviewer #3

(Remarks to the Author)

Pratama et al. present a very thorough exploration of groundwater viruses, their host dynamics, and their contributions to the

overall ecology and specific functions of the environment surveyed. In general, I find the study to be sound and accurately depicted. I have some concerns that will need to be addressed before the work is suitable for publication.

Major concerns

- 1) How many MAGs did you start with before dRep? A concern is that dRep on your MAGs may mean you have lost important CRISPR array variations, as these regions are subject to rapid change based on local exposures, which would not be expected to be stable across sampling site – choosing one representative bin per lineage will flatten this data to only the representative genome's CRISPR complement. Similarly, clustering to vOTUs at 95% ID will also have reduced the genomic diversity of your viral data at a threshold that will obscure specific avoidance mutations within viruses to escape CRISPR targeting (where a 2-mismatch in 50 bp is frequently considered sufficient). I don't think you need to go back to initial MAGs or unclustered viruses, but you will need to position your data as a likely underestimate of connections for your chosen methodology (which relies in large part on CRISPR-target linkages).
- 2) I cannot connect the numbers of samples to the number of wells or years, nor can I link the metatranscriptomic data to these other data. Please include a table clearly defining each sample, ideally as a table in the supplement pdf rather than another stand-alone excel sheet.
- 3) Line 121-130 – this section misses the work by Wu et al (August 2024, <https://www.nature.com/articles/s41467-024-51230-y>, and references within) developing a much larger catalogue of viral genomes from groundwater – I do think this needs to be updated.
- 4) Line 588-592 – no detail is provided on how physicochemical data was measured.
- 5) Why work with three different thresholds of MAG quality? It is not made explicit in the text where these different MAG grouping were used.

Minor concerns

Line 101 – 0.58% for each of these viral phyla, or for them combined (not clear from phrasing), same for line 103, 86.76% belonging to 2 orders – please distinguish between these (i.e., provide the stat for each). Throughout this section, not clear why abundances are being presented as lumped binaries.

Line 115 – please report the % of viruses in Well H41 that were active, to complement the relative transcript expression stat, and to anchor it in the preceding and following sentences, which both discuss % active viruses.

Line 361 – I cannot resolve the stat in the text “while 58.83% of the groundwater AMG were also found with those found in the 20-years of freshwater lake virome study” with the venn diagram in Figure S24b – overlap is much less than 50%, and the numbers don't match either.

Line 471 – missing citation for previous study

Line 472-474 – were the 0.2 μm and 0.1 μm filters in series? Or was 50 L filtered through the 0.2 and 100 L filtered through the 0.1? As written this part of the methods is not clear.

Line 477 – are the stats for 2019 and 2022 metagenomes combined? Awkwardly connected to the two different sequencer details. In general, more detail on sample handling, phenol chloroform extraction method, and sequencing (library prep, insert size, etc.) are needed.

Line 484- was any adaptation to the DNA extraction protocol used to target high molecular weight DNA?

Line 524 – what was the rationale for introducing a second assembly of the same reads (MegaHit) specifically for the viral work?

Line 547 – transposonses – I think this should be transposons

Line 626-637 – these are results, not methods, and need to be more clearly described. Not currently clear what the $n=##$ reporting means.

Line 651 –sum of coverage depth per base pair – sum would not make sense for length disparity concerns between viral and host genome. Do you mean average?

Line 654 – this sentence is results, and also seems out of place here.

Figure 2 – It would be stronger to apply the same well-specific color scheme in A to panels B and C. Same note for Figure S9a and figure S10, S13 – you have two different color schemes for the wells, better to use one. For S13, the more colorful schema would work better.

Figure 2, Supplemental Figure S13 – “macrodiversity” has multiple meanings in ecology, suggest here and in text using Shannon's H index in its place.

Supplemental data

Supplemental dataset names are unintelligible from the file names (e.g., Table S1 is "604347_0_data_set_10652545_snsxvn") – this makes the data very difficult to navigate. Even if on a publication's page the link is named for the table, the user would still receive these unintelligibly named files.

Supplemental figure S5b – why is this merged, rather than by well as in S5a?

Supplemental figure 6a – both parts of the pie chart are labeled "unassigned"

Supplemental Figure S13 – why treat wells H32 and H43 differently (re: Spearman). If you cannot apply the same test, label the panels with the different test explicitly, not just in the legend.

Throughout figure legends, the phrase "All pairwise comparisons shown were statistically significant" is used incorrectly – many non-significant pairs are shown (i.e., displayed), and in some cases p-values reported for them. Please rephrase where relevant to "Statistically significant pairwise comparisons are highlighted with connecting bars, ..."

Figure S16 – does 'highest metatranscriptomic expression above 1%' mean you took the highest transcript abundance for a MAG as their value in this figure? An average, or transcripts per 10,000 bp, or similar, would be a more accurate comparison.

Supplemental Figure S20 – what are these plots sorted/arranged by? Dataset-wide phylum level abundances? It seems arbitrary and is difficult to parse.

Version 1:

Reviewer comments:

Reviewer #1

(Remarks to the Author)

Most of my comments have been adequately addressed. I appreciate the additional statements about caveats, limitations, and uncertainties -- since this paper is based solely on sequencing data. I think the structure of the manuscript is improved and it is more readable.

Reviewer #2

(Remarks to the Author)

Overall Recommendation:

The authors have been largely responsive to the previous round of review, and the manuscript has improved significantly. I believe it has the potential to be suitable for publication in Nature Communications pending satisfactory revision of the following major points.

Major Concerns:

1. Evidence for a Tripartite Relationship:

The section proposing a "Potential tripartite relationship" is currently speculative and requires substantial strengthening. The conclusion is drawn primarily from relative abundance correlations, which is insufficient to demonstrate a direct or functional relationship. The use of the term "episymbiotic" is also premature without experimental or physiological evidence confirming a symbiotic lifestyle. I recommend the authors either:

Tone down the language to highlight this as a hypothetical model for future testing, or incorporate additional, more direct lines of evidence (e.g., proximity ligation data like Hi-C, or genomic evidence for interaction such as metabolite exchange potential).

Furthermore, this model raises an important biological question: what is the proposed role of the viruses? The manuscript should discuss whether these viruses are predicted to target the episymbiotic prokaryotes, the host, or both, and the potential implications of each scenario.

2. Analysis of RNA Viruses in Metatranscriptomic Data:

The availability of high-quality metatranscriptomes presents a valuable opportunity to profile the active RNA virosphere. The analysis appears to be limited to DNA viruses. I strongly encourage the authors to extend their analysis to include RNA viruses. A discussion on the composition and potential ecological roles of these RNA viruses (e.g., as pathogens of eukaryotes or prokaryotes) would greatly enrich the study and provide a more complete picture of the viral community.

Reviewer #3

(Remarks to the Author)

Thank you for your clear and thorough revisions in response to my comments. I have no further comments, and commend

you for an interesting piece of research.

Version 2:

Reviewer comments:

Reviewer #2

(Remarks to the Author)

Thank you very much for the kind response on my concerns. I have no further comments and commend you for an interesting piece of research.

REVIEWER COMMENTS

Reviewer #1 (Remarks to the Author):

This manuscript presents an analysis (and new metagenomic data) focused on groundwater viruses. The system is interesting and the authors attempt to compare it with other systems to put it in context. There is a large focus on tiny prokaryotes (CPR and DPANN) that are prevalent in groundwater. These are all topics of timely interest to the community. However there is a lot of speculation and spinning threads based on limited evidence, which should be toned down significantly.

Response:

We thank the reviewer for highlighting the relevance of our study and for the helpful suggestion to better separate speculation from evidence.

Major findings

- Identified ~257k vOTUs (dereplicated viral contigs >5kb) in groundwater, far more than previous groundwater datasets; mostly Caudoviricetes though 30% unassignable to phylum, with a high degree of novelty (i.e. never before detected in other datasets; but see below for comments on the approach)
- Mapped some (a small fraction, ~11%) of vOTUs to predicted hosts; some mapped to CPR and DPANN organisms
- Estimated virus:host ratios via read mapping to MAGs and vOTUs, found variation across taxa (but see comments below)
- Identified AMGs in these vOTUs related to biogeochemical (C,N,S) cycling
- Found varying degrees of viral overlap between sampling wells, reflecting aquifer hydrology and environmental conditions (esp oxygen)

Strengths

- large dataset of groundwater metagenomes (a poorly characterized habitat), specifically analyzed for viruses. Includes 0.5Tb of existing MG data plus 0.7Tb new MG data, plus 22Gb existing metatranscriptome data
- Good motivation w.r.t. biogeochemistry, prevalence of tiny cells, extremely oligotrophic environment dominated by chemolithoautotrophy
- discovery of new viruses predicted to infect taxa such as CPR, which are poorly characterized
- Introduction is generally well written and motivates the study

Weaknesses

very weak evidence to support some threads in the manuscript (1) novelty, (2) virus decoys, (3) tripartite interactions, (4) influence of AMGs on the system.

Response:

We appreciate the reviewer's summary of the main findings and strengths of our study, and for pointing out areas where our interpretations required more caution. These comments were all deeply considered and have helped us to improve the clarity and balance of the manuscript with specifics provided below.

1) database comparison to assess novelty: The authors found no vOTU overlap between their groundwater dataset and several other viral datasets from various environments. it is impossible to assess whether groundwater viruses are especially novel without some context. What if you used the same procedure to compare marine and freshwater? or two different freshwater datasets? If there is no overlap in these other comparisons either, then the groundwater novelty is unremarkable (by this approach).

Response:

Thank you for this comment. The comparison of our groundwater virome to viromes from other datasets and biomes was performed at a later stage. We agree that providing more context at this point would strengthen its interpretation. As suggested, we applied the same procedure (tools,

thresholds, and clustering approach) not only to compare groundwater with marine and freshwater viromes, but also to perform comparisons within IMG/VR freshwater datasets (e.g., reservoirs, rivers, creeks, lentic and lotic systems). These analyses consistently revealed low vOTU-level overlap between biome types, and even between different freshwater systems. Given this, we reframed this section as ‘comparison of groundwater viruses to other viruses in public databases and across biomes’ rather than over-interpreting the lack of overlap as being uniquely novel. We have also toned down the novelty-focused language throughout. Additionally, we now emphasize that low vOTU overlap appears to be a broader feature of biome-specific viral communities, rather than an observation unique to our groundwater dataset (lines 139-142).

Importantly, we complement this clustering-based approach with a gene-sharing network taxonomic analysis using vConTACT3 (RefSeq v230) and geNomad (gene marker), which further shows that the majority of our vOTUs are unclassified relative to reference databases (lines 118-127).

Suggesting groundwater viruses are underrepresented in current reference databases, and helps contextualize their apparent distinctiveness beyond clustering patterns alone.

2) emphasis on the results from host predictions, (i) but only 11% of the dataset could be matched to a predicted host. Therefore it is misleading to extrapolate from the 11% to the whole system. (ii) The section about virus decoys and alternative genetic codes (lines 280-303) is supported by very little evidence in the current paper and is extremely speculative. Alternative explanations are not explored (e.g. very low burst size in CPR, lots of unidentified CPR viruses in the 89% of vOTUs that could not be matched to a host).

Response:

Thank you for pointing this out. We have removed the speculative argument from ‘Groundwater viruses are predicted to infect diverse prokaryotic phyla’ sections. And overall have toned down speculative interpretations and ensured that claims and conclusions are data-driven.

- A. We agree that host predictions are limited to the subset of vOTUs (~11%) for which confident host linkages could be made. In the revised manuscript, we have carefully reviewed the manuscript to mention this subset of the linkages (~11%) at the start of each relevant section discussing host-based findings (such as line 244 and line 292).
- B. We have substantially restructured and toned down the framing. We also added some limitations of the current virus-host methods and emphasize the need for experimental validation and avoided overstating the results (lines 253-265). We also added alternative explanations for the observed low virus–host ratios and virus–host associations in CPR/DPANN lineages and systematically explain the potential of tripartite interactions (lines 285-331). Specifically, we now discuss that:
 - (i) *In silico* host prediction (lines 291-299) and co-occurrence (lines 300-313).
 - (ii) low burst size (line 320), due to CPR’s minimal biosynthetic capacity, could limit viral replication;
 - (ii) prevalent lysogeny CPR/DPANN viruses may lead to underestimation of active infections (line 321);
 - (iii) physical constraints of episymbiosis might restrict access by free viruses (lines 320-321);
 - (iv) genome streamlining in CPR/DPANN may reduce recognition sites for virus attachment or infection (line 321).
 - (v) A significant fraction of viruses infecting these lineages may reside among the ~89% of unassigned vOTUs, escaping detection due to limitations in current host prediction approaches (line 258-260).

We also revised the text to more clearly explain how different lines of evidence, host prediction (lines 291-299), co-occurrence analysis (lines 300-313), prophage detection (lines 313-316), VHRs

(lines 317-324) and CRISPR spacer matching (lines 323-331), were used to evaluate the possibility of tripartite associations. We now frame this as an exploratory analysis that generates hypotheses for future experimental testing, rather than a conclusion (discussed in the next response).

3) no evidence of tripartite interactions. This whole section and fig 3d-e should be removed. The authors attempt to identify CPR/DPANN host associations via co-occurrence network analysis. However the co-occurring taxa need not be hosts for the CPR/DPANN -- more likely, they are simply co-Responseing to similar environmental drivers. E.g. chloroflexi and nitrifiers also like dark, low nutrient environments. There is no evidence they are physically associated with CPR/DPANN. Therefore, there is no evidence that viruses are somehow involved in a tripartite interaction here. This section is pure speculation.

Response:

We appreciate the reviewer's critical evaluation of this section and agree that co-occurrence alone is not sufficient to infer direct host-symbiont-virus interactions. However, we respectfully disagree that this section and the associated figure should be removed. Our goal in this section was not to claim direct physical interactions nor definitive tripartite relationships but to explore whether multiple independent analyses converge to support a plausible ecological scenario that merits further investigation. We also point out that Reviewer 2 emphasized this section as a particularly intriguing and thought-provoking contribution, which we believe further justifies its inclusion, though, given concerns about it, we have revised considerably.

To strengthen this section and address the concerns raised, we have made the following key revisions:

- A. Clarified our framing: we now explicitly write the section more systematically, from setting up the session and discussing the individual analysis.
- B. Expanded the analytical basis beyond co-occurrence:
 - (i) Combination of co-occurrence and *in silico* virus-host predictions linking vOTUs to CPR/DPANN and other prokaryotes (those with >90% confidence score). Our analysis revealed that 36.6% (467/1,275) of MAGs were associated with other MAGs (Fig. 3d), of which 212 CPR MAGs were associated with non-CPR bacteria and 14 DPANN MAGs were associated with other non-DPANN archaea. Of the MAGs with such associations, 74.7% (349/467) were also linked to vOTUs, supporting the possibility of concurrent host-symbiont-virus interactions (lines 291-299; lines 300-313).
 - (ii) Detection of putative prophages in CPR/DPANN MAGs (106/519 CPR MAGs; 38/131 DPANN MAGs) indicates possible past or ongoing virus-host integration events (Fig. S21) (lines 313-316).
 - (iii) Virus-host ratio (VHR) analysis showed that CPR lineages, despite being highly abundant and active, exhibited lower VHRs than other taxa, suggesting potentially constrained replication dynamics (e.g., low burst size, episymbiosis, lysogeny) (lines 320-324).
 - (iv) CRISPR-spacer matching revealed limited overlap across Patescibacteria and their possible microbial associates, although a few shared matches were detected (Table S10). We found two vOTUs spacers linked to Desulfobacteriota and Patescibacteria (Paceibacteria), but no vOTU spacers linked to archaea and DPANN (lines 324-326).
- C. Revised figure and text accordingly: Fig. 3d-e and Fig. S22 were retained, but the accompanying narrative has been substantially rewritten to reflect a more balanced and cautious interpretation (see Results section lines 300-313).
- D. Lastly, in the revised text, we have 'toned down' our claim and added:
 - "cautiously recognized that co-occurrence alone does not establish direct interactions, as shared environmental preferences may drive similar patterns. Still, as one line of evidence, it provides testable hypotheses for future experimental evaluations once new targeted toolkits become available" (lines 303-305).

- “Collectively, these analyses point to potential tripartite interactions among episyntrophic prokaryotes, their hosts, and viruses. While our results highlight patterns consistent with such relationships, confirmation will require experimental validation or higher-resolution genomic data (for example, CPR/DPANN model system, isolates, single-cell or long-read sequencing)” (lines 328-331).

We believe that removing this section entirely would overlook a potentially important emerging area of research and deprive the field of a framework for future experimental validation.

4) AMG of CPR/DPANN section has a lot of redundancies and speculation, based on limited evidence. line 415 claims "increased presence of AMGs" but relative to what? It is not demonstrated that CPR or groundwater have more AMGs. Likewise "prevalence of AMGs related to carbon and energy" and the comparison to deep sea where there is "an enrichment of AMGs involved in central carbon metabolism" is not supported. Need statistics to show enrichment (and enriched compared to what?).

Response:

Thank you for this important comment. We have substantially revised this section to remove redundancies, tone down the claims, and eliminate speculative language (lines 372-391). In the revised text, we now clearly state the core findings:

- 7.1% of viruses predicted to infect CPR encode AMGs involved in core carbon metabolism (e.g., glycolysis, pentose phosphate pathway) and amino acid biosynthesis.
- 5.8% of viruses linked to DPANN encode AMGs related to pyruvate metabolism, sulfur reduction, and organic nitrogen processing.
- Some AMGs related to the cell surface, while not directly involved in hosts' metabolisms, may influence host-virus interactions, such as attachment.

5) Lots of incomplete sentences and typos in text and figure legends, esp in Methods

Response:

Thank you for spotting the errors. We have carefully read through the materials & methods, figure legends, and adjusted them accordingly.

6) Too long and repetitive in many places

Response:

We thank the reviewer for this helpful observation. We have carefully revised the manuscript to reduce repetition and improve conciseness.

Specific comments:

7) lines 31-33: combine 1st very short paragraph with subsequent. Suggest not focusing the first sentence of the m.s. on "phages" if you also discuss viruses of Archaea.

Response:

We appreciate the reviewer's suggestion and agree that the framing should reflect the broader range of prokaryotic viruses studied. We have revised the first paragraph accordingly for improved clarity (lines 32-34), and discussion on Archaea viruses (lines 63-65).

8) Intro: remove carrot symbols for exponents

Response:

We have removed the caret (^) and replaced them with proper superscript formatting for exponents (e.g., 10³¹). These were initially used to avoid potential confusion with superscripted reference numbers in the text, but we recognize this is non-standard formatting and have corrected it throughout the manuscript (lines 32-36).

9) line 52: "rates comparable to those in sunlit oceanic water" -- do you mean comparable to rates of

chemolithoautotrophy in the surface (which are probably very low), or rates of photoautotrophic C fixation (which are much higher)?

Response:

Thank you for this clarification request. Our reference to “rates comparable to those in sunlit oceanic water” was based on Overholt et al. (2022): <https://doi.org/10.1038/s41561-022-00968-5>, who reported that carbon fixation by chemolithoautotrophs in groundwater surprisingly overlaps with reported rates of photoautotrophic carbon fixation in oligotrophic marine surface waters. We recognize that the original phrasing may lead to ambiguity and have revised the sentence for clarity (lines 57-59).

10) line 66: "viral ecogenomic"?

Response:

Thank you for the suggestion. We clarified the text “viral ecogenomics” (we applied viral ecogenomic analyses that integrate ecological and genomic data to explore viral community structure and function) (lines 72-73).

11) line 71: what do you mean by "selectively infecting"? Of course the viruses have to be specific for the taxa present

Response:

We agree that host specificity is a fundamental property of viruses. We have revised the phrasing to clarify our hypothesis (line 78).

12) line 72 and elsewhere in text and figure legends: remove "pristine". Ill defined and subjective description (there are probably PFAS in the groundwater...)

Response:

We have removed the word pristine. We then explained more about our groundwater system in the introduction (lines 73-77), and more in earlier sections (lines 157-163).

13) line 70-73: I'm not sure what the difference is between the two hypotheses/proposals ("We hypothesize" and "we further propose").

Response:

Thank you for your comment. To improve clarity and avoid redundancy, we merged the original two-part hypothesis into a single statement (lines 77-79). “We hypothesize that viruses shape groundwater microbiomes by infecting ecologically important microbial taxa and contributing novel functions, reflecting adaptations to the subsurface environments”.

14) Fig 1 legend part b: "with the numbers representing sampling replicates" -- which numbers are you referring to?

Response:

Thank you for catching this error. We have adjusted the caption accordingly.

15) Methods - Sampling, lines 472-3: "A total of 50 and 100L..." it is not clear how this was sampled. The legend for fig 1 says sequential filtering, which implies that the same volume would be on both filters. was more volume collected on 0.1? Were there triplicate 0.1 and 0.2 filters for each well?

Response:

We clarified this in the sampling subsection of the materials and methods (lines 427-433). Additionally, we now explicitly refer to Supplementary Table 10, which provides a complete summary of sample numbers, year of sampling and filter types per site, both for metagenomics and metatranscriptomics.

16) How was the 0.1µm filter used vs the 0.2? Did you directly compare the results from the two size fractions? It would be interesting to know if the 0.1 captured different viruses, and your reasoning for sampling both sizes. (there is only a brief mention in line 190 and ordination plots).

Response:

Thank you for this thoughtful comment. The 0.1 µm and 0.2 µm filters were used sequentially during sampling (line 331), as we wanted to capture ultra-small prokaryotes such as CPR bacteria, which are known to pass through 0.2 µm filters.

To evaluate virus community grouping based on the filter-size, we conducted (i) ordination-based analyses, including PCoA (Fig. 2a), NMDS (Fig. S10a) and (ii) compositional analyses (Fig. S9d-f). These analyses showed that filter size explained a small fraction of variation in community structure (PERMANOVA $R^2 = 0.03$, $p = 0.01$), compared to variation across wells ($R^2 = 0.32$) and sampling years ($R^2 = 0.10$). Viral communities were broadly similar between filters, differing mainly in relative abundances.

Notably, many CPR/DPANN MAGs and their predicted viruses were enriched in the 0.1 µm fraction (e.g., Fig. S9, S14d), consistent with the expectation that this filter captures ultra-small lineages and their associated viruses. This provides an independent line of evidence, based on relative abundance patterns, supporting the ecological relevance of including the 0.1 µm fraction. Given the limited overall explanatory power of filter size and the stronger structuring by well and year, we focused our primary ecological analyses at the well level, where patterns were more robust and biologically meaningful.

17) Fig S1 workflow: in "Filtering 3" step, it says "specifically those >100kb". but the input and output from that step are >5kb...

Response:

Thank you for pointing out this confusion. We have clarified the wording in Fig. S1 to make it explicit that additional filtering for mobile elements and genomic islands was performed only on contigs ≥ 100 kb from the full dataset (≥ 5 kb) vOTUs.

Just to be clear, before this step, we also applied a few other filters,

- Step1: recommended cutoffs, threshold and SOP from the individual tools.
- Step2: manually spot-checked contigs from DeepVirFinder, VIBRANT, and geNomad (you may ask why not vs2? because we have also applied SOP to VS2). Specifically, we retain virus contigs meeting one or more of the following: (i) > 0 viral genes, (ii) virus gene = 0 and host gene = 0, (iii) virus contigs length ≥ 1 kb, and (iv) virus contigs with genes $\geq 75\%$ unknown genes. We did another manual check after this.

Additional steps: We trimmed host-associated regions using the CheckV "end-to-end" function and default settings (see details, lines S16-S46).

18) Methods: there are many incomplete sentences throughout, and other grammatical errors, and sentences that say "I did x..." -- Were the methods carefully read and edited by all authors?

Response:

Thank you for this important comment. We carefully read through the materials and methods and made adjustments accordingly.

19) citations: there is some sloppy referencing in the text. e.g. (i) line 112-3: are refs 13 and 19 for the datasets? unclear how they relate. (ii) Line 116-118: are there other refs you can cite besides permafrost/arctic --e.g. marine or freshwater? (iii) refs 49-50 are gut, unrelated (they seem to be supporting the definitions of core, common, and rare -- but these are intuitive and don't need citations). (iv) ref 52

(OMZ) seems unnecessary and potentially misleading, unless you are also going to mention other oxic and anoxic systems (not just the ones published by your group). (v) Ref 53 also seems unnecessary, unless you are going to cite other examples of environmental factors influencing viral communities. (vi) Line 271-3: again seems like a cherry picked result and does not contribute to the discussion at hand; (vii) please remove refs 34-35 here or include a broader discussion of VHRs across systems. (viii) Ref 74 is unnecessary, random, and out of place (line 409). (ix) ref 90 (line 517) is unnecessary (normalization is standard). (x) Ref 112 is unnecessary.

Response:

Thank you for the comments. (i) We have cited the appropriate citation (line 144); (ii) these permafrost and arctic soil studies are a handful of studies with quantitative values (%) of active viruses. Therefore, we keep these studies. However, as the reviewer suggested, we have now added relevant ecosystems, i.e. studies from freshwater and ocean (with a caveat, these studies didn't show any quantification of the active viruses) (lines 149-157). (iii) Ref.49-50, we have removed the citation (line 170). (iv) Ref.52, We have now removed the citation (we reworked the whole paragraph; line 179-182). (v) We removed the reference and reworked this section (line 202). (vi) We changed to a more relevant study from the lake ecosystem (line 280-282). (vii) Ref.34-35, in addition to a more relevant study from the lake ecosystem (line 280-282), we have also added a commentary paper on the broader VHRs across ecosystems (lines 282-284). (viii) Ref.74, We have removed and substantially restructured this whole section (lines 374-391). (ix) Ref.90, We removed this reference (line 553). (xii) Ref.112, We removed this reference (line 711).

20) Fig S6: is gray active/black inactive? need legend

Response:

Thank you for the comment. Fig. S8a illustrates all active viruses, and the pie chart in Fig. S8a shows all active viruses with and without assigned hosts. We have added short titles per figure panels and emphasis "All active virus" for the pie chart.

21) line 118-120: host predictions come later, so perhaps move this sentence

Response:

We have moved the sentence to the host prediction section (now lines 252-253).

22) line 126-130: what's the difference between the Global Ocean Virus db and GOV - c.f. the 0.4-fold and 2.8-fold statistics

Response:

We thank the reviewer for pointing out this potential confusion. The two comparisons refer to different versions of the Global Ocean Virome dataset: (i) the Global Ocean Virome Database version 2 (GOV2) (Ref. 4), and (ii) the updated GOV (GOV2.2) dataset reported by Tian et al. (2024) (Ref. 41). We have now clarified the versioning of these datasets in the main text to avoid confusion. In addition to these two GOV dataset versions and other biomes, our intention was not to cherry-pick values but to acknowledge both the established GOV datasets for context (lines 114-116).

23) line 134-136: was the initial comparison with IMG-VR limited to groundwater? specify here; because later you expand to IMG-VR-freshwater

Response:

Thank you for raising this important point. We have adjusted the text accordingly (line 130).

24) line 146-152: remove this paragraph; all you need is a concluding sentence in the previous paragraph summarizing the findings. Ref 47 is not relevant here, and well-specific patterns are discussed in the following section.

Response:

We removed the paragraph and added a concise summary of the findings in the previous paragraph (lines 139-142).

25) line 162-4: incomplete sentence

Response:

Thank you for catching this error. We revised the sentence accordingly (lines 169-173).

26) line 170-173: remove host prediction sentence, because that comes in a later section

Response:

We removed the host prediction sentence.

27) line 215-217: remove this sentence; the data don't actually show this, and the sentence is unnecessary.

Response:

We removed the sentence as suggested.

28) fig S14b: it's really hard to distinguish the C/O/F patterns. Why not show a tree instead?

Response:

Thank you for your suggestion. We agree that the original figure may have been difficult to interpret due to the complexity of distinguishing class/order/family (C/O/F) patterns. Our intention with Fig. S14b was not to present phylogenetic relationships or detailed taxonomic hierarchies, but rather to provide an overview of the number of MAGs across taxonomic groups. To improve clarity and better serve this purpose, we have simplified the figure to display only the number of MAGs at the phylum level (new Fig. S14b).

29) section starting line 219: earlier you found oxygen to be important; can you tie that in here? how does oxygen relate to these parameters? Concluding sentence is weak, can you revise (or remove)?

Response:

We have revised the section to explicitly address oxygen's role (lines 215-225) and improved the conclusion (lines 225-227).

30) fig 3a: unclear what is being plotted here. Is it the rel abund of these prokaryotes, or the rel abund of predicted hosts? i.e. was the overall abund of Proteobacteria in H14 75%? or, of the hosts predicted in H14, 75% were proteobacteria?

Response:

Thank you for pointing this out. Fig. 3a displays the relative abundance of predicted host phyla only, not the entire prokaryotic community. We have clarified this in both the figure, its caption and the results text to prevent confusion. The relative abundance of all the prokaryotes can be seen in Fig. S14d.

31) host prediction section: The authors should emphasize that this whole discussion is based on just 11% of the vOTUs. (i) So it is not fair to say that Proteobacteria are the most common host in this system. likewise for VHR discussion: (ii) please be explicit about the limitations. What fraction of the data are we actually able to quantify here? What fraction of the reads are represented by the host MAGs and vOTUs with host predictions? How might the conclusions change if the missing 89% of vOTUs infect CPR or DPANN or Proteobacteria (or aren't viruses at all)?

Response:

We agree that these host-virus patterns are based on the subset of vOTUs (~11%) for which host predictions were possible, and we have now mentioned this earlier in the section (line 244; line 293), and did some of these adjustments:

(i) We have revised the text to more clearly state the fraction of the dataset represented (line 244) and to explicitly note this as a conservative estimate (253-260).

(ii) The limitation of the approach used for host prediction is now explicitly stated in the text. We have also emphasized that conclusions about virus-host interactions are framed in this context, and outcomes could shift as host predictions improve or more MAGs become available (253-260). Additionally, we now provide statistics on what proportion of metagenomic reads are represented by the MAGs and the linked vOTUs (Table S5, 'Summary read mapping' tab).

32) line 249-250: SAR11 is Proteobacteria, so that doesn't contrast with your findings. An alternative explanation is that Proteobacteria and their phages are well represented in the dbs, and therefore they are more commonly identified.

Response:

Sorry for the confusion. Our comparison aimed to highlight a key ecological distinction: in marine systems, viruses tend to infect the most abundant host taxa, such as SAR11, whereas in our groundwater system, the common predicted host, i.e., Proteobacteria, was not the most abundant microbial groups. This contrast is not at the phylum level but rather reflects different infection patterns relative to host abundance between the ecosystems. We have reframed this point in the revised manuscript (lines 244-253). Additionally, in response to the suggestion that Proteobacteria and their viruses may be overrepresented in reference databases, potentially biasing host predictions, we have now included a statement acknowledging this as an alternative explanation (lines 265-266).

33) line 267: remove statement about "higher viral diversity" - the VHR does not specifically address viral diversity

Response:

We revised the sentence accordingly (line 276).

34) line 276-7: clarify that 18.42% is of host-matched vOTUs (only 11%)

Response:

To avoid confusion, we now specify that the 18.4% of vOTUs linked to Patescibacteria refers only to the subset of vOTUs with host predictions (11.6% of total vOTUs), not the full dataset (line 293).

35) line 312-321: this describes host predictions, which are discussed in the previous section. Move these results to the previous section (where you already mention the same results, i.e. n=1771 CPR-linked vOTUs)

Response:

We moved the earlier sentence and revised the text: "We identified virus-host associations (at a high host confidence score $\geq 90\%$) for 11.6% (9,615) of vOTUs, linking them to microbial MAGs spanning 60 phyla, 143 classes, 339 orders, and 650 families (Fig. S18 and Table S8). Among these identified associations (11.6%), Proteobacteria accounted for the highest proportion of virus-host linkages (21%), followed by Patescibacteria (18%)" (lines 241-243). However, we maintained in silico host prediction to emphasize the virus linkages for CPR and DPANN (lines 292-300).

36) line 371-372: please explain what the values mean ("45.6% relative abundance" - what is the numerator and denominator?)

Response:

The values refer to the relative expression of viral AMGs. The percentages represent the normalized expression per AMG category. We have revised the text (lines 350-351).

37) line 394-6: combine with previous paragraph and tone down the conclusion. I do not agree that "compelling evidence" has been presented for influencing metabolism, nutrient cycling, and "ecosystem adaptation" (whatever that is)

Response:

We have now merged it into the final paragraph and removed the phrase "compelling evidence" (lines 369-371).

38) line 398: AMG's in viruses of CPR and DPANN (AMG's are not in bacteria/archaea, they are in viruses)

Response:

Correct, we have revised the section title to clarify that the auxiliary metabolic genes (AMGs) are virally encoded and are associated with predicted CPR and DPANN hosts (line 373).

39) line 401-402: how does this relate to CPR metabolism? CPRs are fermentative anaerobes

Response: g

Good point, we added some examples to be more specific. Please find the modified text below: Among CPR-associated viruses, 7.1% (125 vOTUs) encoded auxiliary metabolic genes (AMGs) related to central carbon metabolism (e.g., glycolysis, pentose phosphate pathway), amino acid biosynthesis, and nutrient transport (Fig. S26, Table S12). These AMGs may complement incomplete host pathways and help overcome metabolic bottlenecks. For example, several vOTUs encoded *phosphoglucose isomerase*, an enzyme that interconverts glucose-6-phosphate and fructose-6-phosphate and is sometimes missing or only partially encoded in CPR genomes. By restoring or enhancing glycolytic flux, such AMGs may increase ATP production via substrate-level phosphorylation, thereby boosting the host's fermentative capacity under energy-limited conditions and supporting viral replication. Similarly, among viruses predicted to infect DPANN archaea, 5.8% (105 vOTUs) carried AMGs involved in carbon processing (e.g., pyruvate metabolism), sulfur reduction, and organic nitrogen turnover. Enzymes such as pyruvate dehydrogenase and threonine/serine dehydratases could redirect host carbon flux toward pathways that favor viral replication, even at the expense of host energy conservation. Additionally, some AMGs related to cell surface, while not directly involved in hosts' metabolisms, may influence host-virus interactions, such as attachment (lines 373-392).

40) line 406: what is the point about chaperones and cell surface genes? how does that relate to metabolism?

Response:

We agree that chaperone and surface-associated AMGs are not directly involved in core metabolic pathways. We have revised the sentence accordingly. We have also substantially revised this section (lines 373-392).

41) line 407-410: remove; does not fit with the narrative and is speculative.

Response:

Thank you for the comment. We removed the redundancies and speculative parts. We have also substantially revised this section (lines 373-392).

42) lines 424-8, 439-44 are redundant with other text. this section could be streamlined substantially.

Response:

Thank you for the feedback. We revised the section to remove repetition and speculative statements, as suggested (lines 373-392).

43) line 435: "19% of virus populations, with 9% observed" -- what do those values mean? 9% observed?

Response:

Thank you for the comment. We clarified this sentence (lines 373-392). To answer the comment, 19% is the estimated true proportion of virus populations carrying AMGs after accounting for

genome fragmentation (using a conversion factor). While ~9% is the directly observed proportion of virus populations carrying AMGs in fragmented metagenomic data (before correction- Tian et al, Ref.31). We have also substantially revised this section (lines 373-392).

44) line 437-8: "suggesting the ecological importance of AMGs are likely important" (grammar)

Response:

Thank you for the comment. We adjusted the text accordingly. We have also substantially revised this section (lines 373-392).

45) line 449: "higher-than-expected" -- based on what? was there a stated hypothesis about the rarity of CPR/DPANN viruses? or just your hunch?

Response:

We appreciate the reviewer's attention to this point. Our original phrasing was imprecise. We removed "higher-than-expected" and now state more precisely that CPR- and DPANN-associated viruses were frequently detected and often encoded AMGs, which is notable given the limited data available from these systems (line 397).

46) line 517-519: 25% of reads mapping to MAGs does not seem very substantial. This is >75% in many aquatic systems. Be careful what you say about the microbial community based on just 25% of it.

Response:

We appreciate the reviewer's concern. The '25%' referenced in our manuscript (line 507-514) refers to the '--min-covered-fraction' parameter that is a minimum 25% genome coverage required to retain a MAG/ only MAGs with $\geq 25\%$ breadth of coverage is analyzed.

Our actual read total recruitment rate (proportion of reads mapping to all MAGs) was 25.9%, comparable with requirement rates in aquatic systems (<https://www.nature.com/articles/s41597-022-01392-5>). For comparison, the requirement rate of MAG in the soil system is <5% (<https://www.nature.com/articles/s41587-020-0718-6>) (summary of the read mapping in Table S5).

47) line 547: "transposonses"

Response:

Thank you for spotting the typo. We have adjusted it accordingly (line 541).

Reviewer #2 (Remarks to the Author):

The study instigates metagenomic and metatranscriptomic data to identify 257,252 viral operational taxonomic units (vOTUs), with 99% representing novel viruses, underscoring the vast uncharted viral diversity in groundwater ecosystems. This aligns with previous studies suggesting that subsurface environments harbor unique viral communities distinct from surface ecosystems. The high novelty of these viruses highlights the need for expanded viromic exploration in subsurface habitats to better understand their ecological roles.

Authors found that viruses in groundwater primarily infect Proteobacteria, Candidate Phyla Radiation (CPR) bacteria, and DPANN archaea—key microbial groups in subsurface ecosystems. A particularly intriguing observation is the low virus-to-host ratio in CPR bacteria, coupled with the presence of viral CRISPR spacers targeting multiple hosts. The authors propose a "virus decoy" mechanism, where CPR bacteria may act as viral sinks, absorbing infection pressure and thereby protecting more metabolically active bacterial hosts. This hypothesis, if validated, could reshape our understanding of viral predation dynamics in energy-limited ecosystems. In addition, a major contribution of this study is the identification of 3,378 vOTUs encoding AMGs involved in carbon, nitrogen, and sulfur cycling. These AMGs suggest that viruses actively reprogram host metabolism, potentially enhancing nutrient turnover in groundwater. Notably, 31.5% of host metabolic modules were targeted by viruses, indicating a substantial viral influence

on microbial metabolic networks. This finding supports the growing recognition of viruses as key players in biogeochemical cycling, extending beyond their traditional role as microbial predators.

Overall, this study advances our understanding of viral ecology in groundwater ecosystems, revealing unprecedented diversity, novel host interactions, and metabolic reprogramming capabilities. The proposed "virus decoy" mechanism and the widespread presence of AMGs underscore the need to integrate viruses into models of subsurface biogeochemical cycling.

Response:

We thank Reviewer #2 for the constructive comments and your recognition of the study's contribution to understanding viral diversity, host interactions, and metabolic potential in groundwater ecosystems. We also appreciate the recognition of the proposed "virus decoy" mechanism and its importance for integrating viruses into subsurface biogeochemical models. We have carefully addressed all suggestions to improve the manuscript.

However, there are several Limitations and Open Questions:

1) Lack of Cultured Representatives

While the study identifies numerous novel viruses, the absence of isolated viral-host pairs limits mechanistic validation of the proposed interactions. Future studies should prioritize cultivation-based approaches to confirm host specificity and infection dynamics, particularly for CPR bacteria and DPANN archaea.

Response:

We thank the reviewer for highlighting the important challenge posed by the lack of cultured model systems for CPR bacteria and DPANN archaea. Indeed, the limited availability of isolated viral-host pairs restricts direct mechanistic validation of viral infection and host specificity in these lineages. Our study leverages metagenomic and bioinformatic approaches to provide the first large-scale insights into virus-host associations and the functional potential of CPR- and DPANN-infecting viruses in groundwater ecosystems. While cultivation-based studies remain essential for detailed mechanistic understanding, the current scarcity of such models underscores the value of cultivation-independent methods for expanding our knowledge of these elusive microbial groups. We fully agree that future efforts combining cultivation, single-cell, and viral tagging approaches will be critical to validate and deepen our findings. We explicitly acknowledge this both in the introduction: "Due to the scarcity of cultured representatives for CPR bacteria, DPANN archaea, and their viruses, cultivation-independent genomic approaches are one mechanism to uncover their biology, including infecting viruses, virus-encoded AMGs, and virus-host interactions" (lines 65-67) and the conclusion in the manuscript (lines 405-421).

2) Limited Comparative Analysis with Other Groundwater Systems

The study focuses on seven wells in the Hainich CZE, which, while valuable, represents a relatively narrow sampling scope. Comparative studies incorporating data from other groundwater ecosystems (e.g., pristine vs. contaminated aquifers, deep vs. shallow groundwater) would strengthen the generalizability of the findings. Additionally, benchmarking against existing groundwater virome datasets could help discern whether the observed viral diversity and host interactions are site-specific or broadly representative.

Response:

We appreciate this important point. While our study focuses on a single location, the Hainich CZE provides a well-characterized and ecologically diverse model system spanning oxic-anoxic, shallow-deep and hydrologically connected-isolated gradients.

To place our findings in a broader context, we compared our viruses against multiple public datasets, and across different biomes (Fig. S5-S6), including other groundwater viromes (e.g., Scandinavian virome study, New Zealand Groundwater Virome, the current IMG/Vr viroec database for groundwater ecosystem, and the recent groundwater virome catalogue), freshwater

entries from IMG/VR, and broader biome databases (e.g., GOV2, GSV, GVD). These comparisons (Fig. S5–S6) revealed low overlap, consistent with viral endemism in groundwater (lines 133–135).

We agree that future comparative studies across additional groundwater systems will be essential for assessing the generality of these patterns. Similar to the response to reviewer #1 (comment No.1): We have also toned down the novelty-focused language throughout the manuscript. We now emphasize that low vOTU overlap appears to be a broader feature of biome-specific viral communities, rather than an observation unique to our groundwater dataset (lines 139–142).

3) Dynamic Viral-Host Relationships Require Further Investigation

The study provides a snapshot of viral diversity, but longitudinal sampling could reveal how viral communities shift in response to seasonal or anthropogenic perturbations. Although the authors employed iPHoP (a machine-learning-based host prediction tool integrating BLAST, CRISPR-spacer matching, oligonucleotide frequency, and protein clustering), host assignment remains probabilistic. Experimental validation (e.g., single-cell genomics or phage isolation) would enhance confidence in the predicted interactions, particularly for the proposed "virus decoy" mechanism.

Response:

We thank the reviewer for highlighting these important limitations. We agree that the absence of cultured representatives and the inherently probabilistic nature of current *in silico* host prediction tools (e.g., iPHoP) limit mechanistic interpretations of virus–host interactions, particularly for CPR and DPANN lineages. We have now explicitly stated this limitation in the revised text and emphasized that our host assignments likely represent conservative estimates based on current best practices (lines 253–265).

In response to the suggestion for experimental validation, we now also highlight that approaches such as single-cell genomics (lines 405–421) and phage isolation would provide important validation of our predicted interactions, including the hypothesized virus decoy mechanism (lines 322–331).

Regarding temporal dynamics, while our study primarily provides a spatial snapshot, we include samples from two different years collected from the same wells. This allowed us to detect year-specific variation in viral communities (Fig. 2a; and Fig. S11f; line 187), providing partial insight into longer-term shifts in viral community structure.

4) Quantifying the Ecological Impact of AMGs

While AMGs suggest metabolic reprogramming, their actual contribution to biogeochemical fluxes remains unclear. Future studies could employ stable isotope probing (SIP) or metabolomics to assess whether viral AMGs directly influence carbon, nitrogen, or sulfur cycling rates. Additionally, transcriptomic and proteomic profiling of infected hosts could help determine the functional expression of these AMGs under *in situ* conditions

Response:

We thank the reviewer for this insightful comment. We agree that directly quantifying the biogeochemical impact of viral AMGs remains a key frontier for the field. In this study, we assessed *in situ* expression of AMGs using metatranscriptomic data sampled from the same site (see Fig. 4), which revealed active transcription of virus-encoded genes linked to carbon, nitrogen, and sulfur cycling across wells. Additionally, we highlight expression patterns of specific AMG categories in Fig. S25.

While metatranscriptomics provides evidence for AMG activity at the RNA level, we agree that future approaches such as stable isotope probing (SIP), metabolomics, and proteomics of infected

hosts might help to directly link viral AMG to elemental fluxes and to assess their impact on host metabolism. However, such technologies are both not readily available outside speciality (often large-scale funded national) labs and will be challenged to differentiate cellular and viral AMG homologs to quantitatively assess virus AMG impacts. Though perhaps now that we have deep first descriptions for virus diversity in these systems, these more advanced and challenging techniques could help focus future hypothesis-testing research. We have clarified these points in the revised discussion (lines 355-357).

Reviewer #3 (Remarks to the Author):

Pratama et al. present a very thorough exploration of groundwater viruses, their host dynamics, and their contributions to the overall ecology and specific functions of the environment surveyed. In general, I find the study to be sound and accurately depicted. I have some concerns that will need to be addressed before the work is suitable for publication.

Response:

We thank the reviewer for their positive assessment of our study and for acknowledging the thoroughness and soundness of our work. We appreciate the constructive feedback and have carefully addressed all concerns to improve the manuscript.

Major concerns

1) How many MAGs did you start with before dRep? A concern is that dRep on your MAGs may mean you have lost important CRISPR array variations, as these regions are subject to rapid change based on local exposures, which would not be expected to be stable across sampling site – choosing one representative bin per lineage will flatten this data to only the representative genome’s CRISPR complement. Similarly, clustering to vOTUs at 95% ID will also have reduced the genomic diversity of your viral data at a threshold that will obscure specific avoidance mutations within viruses to escape CRISPR targeting (where a 2-mismatch in 50 bp is frequently considered sufficient). I don’t think you need to go back to initial MAGs or unclustered viruses, but you will need to position your data as a likely underestimate of connections for your chosen methodology (which relies in large part on CRISPR-target linkages).

Response:

Thank you for this insightful comment. We agree that dereplication of MAGs using dRep and clustering viruses into vOTUs at $\geq 95\%$ ANI can reduce the resolution of CRISPR-based virus–host linkage detection.

In particular, collapsing MAGs likely underrepresents strain-level CRISPR array diversity, which can vary across closely related genomes. Likewise, vOTU clustering may obscure fine-scale viral mutations that affect CRISPR evasion. We chose to use dereplicated MAGs and vOTUs to align with current best practices in host prediction pipelines (e.g., iPHoP), which operate at genus-level resolution and are not computationally scaling to individual genomes that readily exist right now. However, reflecting on this, given the reviewer’s comments, we acknowledge that our method provides a conservative estimate of CRISPR-based virus–host associations. These limitations have been clarified in the revised manuscript (Results and Discussion, lines 253-263; materials and methods, lines 651-653).

2) I cannot connect the numbers of samples to the number of wells or years, nor can I link the metatranscriptomic data to these other data. Please include a table clearly defining each sample, ideally as a table in the supplement pdf rather than another stand-alone excel sheet.

Response:

We agree that clearer documentation of sample metadata will improve transparency and reproducibility. To further assist readers in linking samples to wells, years, and sequencing type, we have added a stand-alone summary table (in Table S1b). This table lists all samples used in the

study, along with their associated metadata, including sample ID, well, filter, sampling year, and data type (metaG or metaT).

3) Line 121-130 – this section misses the work by Wu et al (August 2024, <https://www.nature.com/articles/s41467-024-51230-y>, and references within) developing a much larger catalogue of viral genomes from groundwater – I do think this needs to be updated.

Response:

Thank you for pointing out the recent work by Wu et al. (2024). We agree that this study represents a major contribution to the cataloguing of groundwater viruses and should be cited for context. At the time we completed our analysis, this work was not yet available. However, as the reviewer has requested this, we have now incorporated it into our revised manuscript and also updated analyses to include it where possible at this late stage of our work. Specifically, we have:

1) updated the relevant paragraph to include comparison with the ~108K vOTUs ≥ 10 kb reported in Wu *et al* (lines 111-112).

2) added the data to our comparison analysis, at vOTU level. This can be seen in the results and discussion (lines 128-142) and the supplementary Fig. S5-S6.

3) updated the materials and methods to include the Wu *et al* dataset (section Viral clustering and database comparison; lines 605-630).

4) Line 588-592 – no detail is provided on how physicochemical data was measured.

Response:

The description of physicochemical data was mentioned previously (Kohlhepp et al *Hydrol. Earth Syst. Sci.*, 2017; Lehmann, R, & Totsche, KU. *J Hydrol.* 2020; To be self-contained, we have added a description of the physicochemical measurements to the materials and methods (section ‘Sample collection, DNA extraction, and metagenomic sequencing’; lines 467-477), including the parameters measured, citation and instrumentation or protocols used.

5) Why work with three different thresholds of MAG quality? It is not made explicit in the text where these different MAG grouping were used.

Response:

To clarify, we followed established community standards to classify metagenome-assembled genomes (MAGs) by quality. Specifically, we applied quality score thresholds to identify medium-quality (completeness $>50\%$, contamination $\leq 10\%$) and high-quality (completeness $\geq 90\%$, contamination $\leq 5\%$) MAGs. Only MAGs meeting these medium- or high-quality criteria (totaling 1,275 MAGs) were retained for all subsequent analyses, including virus–host predictions. We have revised the text to make this clearer by repositioning the summary sentence immediately after the filtering criteria (materials and methods, lines 491-496).

Minor concerns

6) Line 101 – 0.58% for each of these viral phyla, or for them combined (not clear from phrasing), same for line 103, 86.76% belonging to 2 orders – please distinguish between these (i.e., provide the stat for each). Throughout this section, not clear why abundances are being presented as lumped binaries.

Response:

We have simplified this section and corrected accordingly (lines 118-127). In addition, we have updated the analysis using the latest version of vConTACT3 v3.1.4 (gene-sharing network) and complement it with geNomad taxonomic assignment (gene markers).

7) Line 115 – please report the % of viruses in Well H41 that were active, to complement the relative transcript expression stat, and to anchor it in the preceding and following sentences, which both discuss % active viruses.

Response:

We have now added the proportion of active viruses identified in Well H41 (line 147).

8) Line 361 – I cannot resolve the stat in the text “while 58.83% of the groundwater AMG were also found with those found in the 20-years of freshwater lake virome study” with the venn diagram in Figure S24b – overlap is much less than 50%, and the numbers don’t match either.

Response:

We appreciate the opportunity to clarify the apparent discrepancy. We doubled-checked and updated the Venn diagrams in Fig. S24b. Now, it depicts overlap at the AMG cluster level, whereas the percentages reported in the main text (58.83% for freshwater) refer to AMG protein-level overlap. Specifically, we compared the 4,093 AMG proteins identified in this study with those in the two external databases and found that 2,408 of them overlapped in the freshwater datasets, respectively. We have clarified this distinction in both the main text (lines 341-342) and the figure (Fig. S24b).

9) Line 471 – missing citation for previous study

Response:

We have added the citation to the previous study (now line 425).

10) Line 472-474 – were the 0.2 µm and 0.1 µm filters in series? Or was 50 L filtered through the 0.2 and 100 L filtered through the 0.1? As written this part of the methods is not clear.

Response:

We agree that the original phrasing was unclear. We have now clarified this point in the materials and methods section (line 431).

11) Line 477 – are the stats for 2019 and 2022 metagenomes combined? Awkwardly connected to the two different sequencer details. In general, more detail on sample handling, phenol chloroform extraction method, and sequencing (library prep, insert size, etc.) are needed.

Response:

We have clarified the sample handling, detailed the phenol:chloroform extraction, library prep and sequencing platform in the materials and methods (lines 427-442). We also added the library prep and sequencing platform information to Supplementary Table 1.

12) Line 484- was any adaptation to the DNA extraction protocol used to target high molecular weight DNA?

Response:

We confirm that the protocol was adopted to preserve high molecular weight DNA (line 433-434).

13) Line 524 – what was the rationale for introducing a second assembly of the same reads (MegaHit) specifically for the viral work?

Response:

The rationale for introducing a second assembly using MEGAHIT was to maximize the recovery of viral contigs. Viral genomes are often underrepresented or fragmented in bulk metagenomic assemblies, and different assemblers use distinct heuristics that can impact viral genome recovery. MEGAHIT has been shown to recover viral sequences that may be missed by other assemblers (Roux et al., 2017, PeerJ 5:e3817), particularly for low-abundance or short genomes. By including MEGAHIT alongside our primary assembly pipeline, we aimed to improve overall vOTU recovery. We have clarified this rationale in the revised manuscript (lines 517-520).

14) Line 547 – transposonses – I think this should be transposons

Response:

We have adjusted it accordingly (similar to Reviewer #1, comment #48; now line 541).

15) Line 626-637 – these are results, not methods, and need to be more clearly described. Not currently clear what the n=## reporting means.

Response:

We agree that this section needed clearer wording. The values reported as “n = XX” refer to the “number” of predicted protein sequences in each database before and after filtering (in other parts ‘n’ referred to ‘numbers’ of samples, AMGs, etc). To clarify this, we removed the use of “n = XX” throughout the manuscript (lines 605-636).

16) Line 651 –sum of coverage depth per base pair – sum would not make sense for length disparity concerns between viral and host genome. Do you mean average?

Response:

Our approach follows the method described in Emerson *et al.* (2018, Nat. Microbiol. <https://www.nature.com/articles/s41564-018-0190-y>), where the average per-base coverage depth (i.e., depth normalized by genome length) is used to compute the virus-host ratio (VHR), normalized by sequencing depth. We have revised the manuscript text to clarify this (lines 647-651).

17) Line 654 – This sentence is a result, and also seems out of place here.

Response:

We agree that this sentence presents a result and have moved it from the methods to the results section (lines 239-241).

18) Figure 2 – It would be stronger to apply the same well-specific color scheme in A to panels B and C. Same note for Figure S9a and figure S10, S13 – you have two different color schemes for the wells, better to use one. For S13, the more colorful schema would work better.

Response:

We agree that consistent colour schemes support the clarity. We have carefully reviewed the figures mentioned. In Figure 2 (b and c), we intentionally used a different colour scheme, which is to reflect their focus on well-level comparisons, rather than showing different layers of samples (year, well, filters) in panel 2a. This colour scheme is applied consistently across other figures (e.g., Fig S9a, S10, S13 and the rest of the figures). For Fig. S13 in particular, we opted to retain the well-specific scheme for consistency with Fig. 2 b-c and Fig. S10, where all analyses are resolved by well assessments. However, we provided diversity analysis across these different groupings: year, and oxic/anoxic wells (Fig. S11).

19) Figure 2, Supplemental Figure S13 – “macrodiversity” has multiple meanings in ecology, suggest here and in text using Shannon’s H index in its place.

Response:

We agree that clarity in terminology is important, especially for metrics like macrodiversity that can have multiple interpretations. In all relevant figures (e.g., Fig. 2, Fig. S13), we explicitly labeled the plots as “**Macrodiversity (Shannon’s H)**” and refer to it this way throughout the text as well. We believe this format clearly communicates both the ecological context (macrodiversity) and the specific index used (Shannon’s H) and thus avoids ambiguity. We also define macrodiversity as early as possible in the main manuscript (lines 183-185). For consistency, we have double-checked all mentions of this metric to ensure the same phrasing is used throughout.

Supplemental data

20) Supplemental dataset names are unintelligible from the file names (e.g., Table S1 is “604347_0_data_set_10652545_snsxvn”) – this makes the data very difficult to navigate. Even if on a

publication's page the link is named for the table, the user would still receive these unintelligibly named files.

Response:

We agree that the automatic renaming of supplementary files by the submission platform may cause confusion. To improve clarity, we have ensured that each supplementary table is clearly labeled within the file name that indicates information contained in (for example, "Supplementary Table 1. Metagenomic samples overview"). These table names are explicitly referenced in the main text on the list of Supplementary Information. These steps should help readers identify and navigate the supplementary materials more easily, regardless of file renaming by the publisher.

21) Supplemental figure S5b – why is this merged, rather than by well as in S5a?

Response:

We have revised the figure and analysis accordingly.

22) Supplemental figure 6a – both parts of the pie chart are labeled “unassigned”

Response:

We have revised the figure accordingly.

23) Supplemental Figure S13 – why treat wells H32 and H43 differently (re: Spearman). If you cannot apply the same test, label the panels with the different test explicitly, not just in the legend. Throughout figure legends, the phrase “All pairwise comparisons shown were statistically significant” is used incorrectly – many non-significant pairs are shown (i.e., displayed), and in some cases p-values reported for them. Please rephrase where relevant to “Statistically significant pairwise comparisons are highlighted with connecting bars, ...”

Response:

Thank you for this helpful observation. We clarify that the choice between Pearson and Spearman correlation in Figure S13 was based on data normality test and consistent with best practices for correlational analyses. For clarity, we have added this information on the figure panel labels in S13 to explicitly state whether Pearson or Spearman correlations were used.

We also thank the reviewer for pointing out the imprecise phrasing. Throughout the relevant figure legends, we have revised the text to state: “Statistically significant pairwise comparisons are highlighted with connecting bars”, instead of the misleading statement “All pairwise comparisons shown were statistically significant.”

We further clarify that only statistically significant pairwise comparisons are marked with connecting bars or asterisks. While overall test *p*-values are sometimes reported in figure panels regardless of significance, non-significant pairwise comparisons are not highlighted in the plots. We hope this revision improves the clarity and interpretation of the figures.

24) Figure S16 – does ‘highest metatranscriptomic expression above 1%’ mean you took the highest transcript abundance for a MAG as their value in this figure? An average, or transcripts per 10,000 bp, or similar, would be a more accurate comparison.

Response:

We thank the reviewer for this comment and understand the confusion around “highest metatranscriptomic expression above 1%.” To clarify, this threshold refers to MAGs whose total normalized relative transcript abundance $\geq 1\%$ and were thus retained for visualization in Fig. S16a. As described in the revised figure legend and materials and methods, transcript abundances were normalized using housekeeping gene *rpoB*-based scaling factors and further corrected for average genome size (materials and methods, lines 691-716).

25) Supplemental Figure S20 – what are these plots sorted/arranged by? Dataset-wide phylum level abundances? It seems arbitrary and is difficult to parse.

Response:

Thank you for this helpful comment. The plots in Fig. S20 are ordered by host phylum, and the phyla are ordered from top to bottom based on their overall virus-to-host abundance ratio (VHR) across the full dataset (as shown in Fig. 4c), from highest to lowest. This consistent ordering across wells was intended to facilitate cross-panel comparisons of lineage-specific patterns. We have clarified this rationale in the revised figure legend.

REVIEWER COMMENTS

Reviewer #1 (Remarks to the Author):

Most of my comments have been adequately addressed. I appreciate the additional statements about caveats, limitations, and uncertainties -- since this paper is based solely on sequencing data. I think the structure of the manuscript is improved and it is more readable.

Response:

We thank the reviewer for their positive feedback on the revised manuscript's improved structure and clarity.

Reviewer #2 (Remarks to the Author):

Overall Recommendation:

The authors have been largely responsive to the previous round of review, and the manuscript has improved significantly. I believe it has the potential to be suitable for publication in Nature Communications pending satisfactory revision of the following major points.

Response:

We thank the reviewer for the positive evaluation and constructive feedback. We will address each of the major points raised below in a point-by-point manner.

Major Concerns:

1. Evidence for a Tripartite Relationship:

The section proposing a "Potential tripartite relationship" is currently speculative and requires substantial strengthening. The conclusion is drawn primarily from relative abundance correlations, which is insufficient to demonstrate a direct or functional relationship. The use of the term "episymbiotic" is also premature without experimental or physiological evidence confirming a symbiotic lifestyle. I recommend the authors either:

Tone down the language to highlight this as a hypothetical model for future testing, or incorporate additional, more direct lines of evidence (e.g., proximity ligation data like Hi-C, or genomic evidence for interaction such as metabolite exchange potential).

Response:

We thank the reviewer for this valuable comment and appreciate the opportunity to clarify this section further. We would like to emphasize that our inference regarding potential multi-partite interactions is not based solely on relative abundance correlations, but instead integrates four independent genome-resolved lines of evidence: (i) virus–host predictions, (ii) MAG co-association patterns, (iii) prophage detection, and (iv) CRISPR spacer targeting. Together, these analyses suggest possible ecological associations, though we fully acknowledge that they do not constitute direct experimental proof of interaction.

A recent paper in *Nature Microbiology*: <https://www.nature.com/articles/s41564-025-02149-7> showed multi-layer nested viromes in DPANN–haloarchaeal systems, and proposed the ecological role of viruses as modulators of host–symbiont interactions. The author used CRISPR screening only for host assignment analysis (no experimental data), which we also provided. However, for rigor, we also added three other independent lines of evidence to create what we felt was a stronger supporting analysis.

As noted in our previous revision, we have already substantially revised this section to clarify the scope and speculative nature of the proposed interactions. In the current version, we have further softened the language to explicitly frame this section as a hypothesis-generating model for future testing, rather than evidence for a demonstrated tripartite relationship. The text now clearly states that the proposed interactions are based on genome-inferred analyses (**lines 292-296**) and should be regarded as working hypotheses (**lines 342-343**).

We also reiterate that the term *'episymbiotic'* is used exclusively for CPR/DPANN–microbe host associations, not for their viruses. To prevent misunderstanding and better reflect the exploratory nature of this section, it has been retitled “Potential multi-layer interactions in the groundwater ecosystem.”

We believe that these additional clarifications and revisions go significantly beyond the current ‘standard’ set in the *Nature Microbiology* paper, and we hope appropriately balance caution and scientific curiosity, addressing the reviewer’s concern while preserving a concept that another reviewers highlighted as an important and interesting aspect of our study.

Furthermore, this model raises an important biological question: what is the proposed role of the viruses? The manuscript should discuss whether these viruses are predicted to target the episymbiotic prokaryotes, the host, or both, and the potential implications of each scenario.

Response:

Good point. We have added a few sentences to address this (**lines 337-340**): “Viruses that target the CPR/DPANN organisms may act as hyperparasites⁶³, controlling their population burden. Conversely, viruses infecting the microbial hosts could indirectly restructure or destabilize the entire partnership. Viruses that dual-target both partners could pose top-down pressure to both partners simultaneously.”

2. Analysis of RNA Viruses in Metatranscriptomic Data:

The availability of high-quality metatranscriptomes presents a valuable opportunity to profile the active RNA virosphere. The analysis appears to be limited to DNA viruses. I strongly encourage the authors to extend their analysis to include RNA viruses. A discussion on the composition and potential ecological roles of these RNA viruses (e.g., as pathogens of eukaryotes or prokaryotes) would greatly enrich the study and provide a more complete picture of the viral community.

Response:

We thank the reviewer for this thoughtful suggestion and fully agree that RNA viruses are an important, yet often overlooked, component of environmental viral communities. We appreciate the reviewer’s perspective that metatranscriptomic data can, in principle, provide valuable insights into RNA viral diversity and activity.

However, we would like to clarify that the metatranscriptomic datasets used here were generated using bulk microbial RNA extraction and library preparation protocols that were not optimized for the recovery or enrichment of RNA viral sequences. Thus, we decided to focus our analysis on dsDNA viruses, where detection sensitivity and genome reconstruction are robustly established through community-vetted SOPs and strongly supported by our data. We agree that a dedicated RNA virome investigation would significantly complement our findings, and we note that such analyses are currently underway by our collaborators.

In response to this helpful comment, we have expanded the discussion to explicitly acknowledge the ecological significance of RNA viruses. The following paragraph has been added to the conclusion (**lines 416–422**):

"Although our analyses focus on dsDNA viruses, RNA viruses are also expected to occur in groundwater ecosystems. Their detection and characterization remain challenging due to methodological constraints and limited reference data, as most studies to date have targeted contamination^{68,77} events rather than ecological processes. Nonetheless, their potential influence on microbial dynamics and biogeochemical carbon cycles should not be overlooked, as demonstrated in other environments, like the ocean¹⁰. Future RNA viromics efforts are essential to characterize this component of the virosphere and integrate it with our DNA-based framework.”

We hope this clarification and add discussion sufficiently address the reviewer's concern while maintaining the methodological integrity and scope of the current study.

Reviewer #3 (Remarks to the Author):

Thank you for your clear and thorough revisions in response to my comments. I have no further comments and commend you for an interesting piece of research.

Response:

We thank the reviewer for the positive feedback and constructive comments, which have helped improving the manuscript.